# Parameters vs. Context: Fine-Grained Control of Knowledge Reliance in Language Models

**Baolong Bi**[1,2]   **Shenghua Liu**[1,2*]   **Yiwei Wang**[3]   **Yilong Xu**[1,2]
**Junfeng Fang**[4]   **Lingrui Mei**[1,2]   **Xueqi Cheng**[1,2]

[1]State Key Laboratory of AI Safety, Institute of Computing Technology, CAS

[2]University of Chinese Academy of Sciences [3]University of California, Merced

[4]National University of Singapore

{bibaolong23z, liushenghua, xuyilong23s, meilingrui25b, cxq}@ict.ac.cn

wangyw.evan@gmail.com, fangjf1997@gmail.com

## ABSTRACT

Retrieval-Augmented Generation (RAG) mitigates hallucinations in Large Language Models (LLMs) by integrating external knowledge. However, conflicts between parametric knowledge and retrieved context pose challenges, particularly when retrieved information is unreliable or the model's internal knowledge is outdated. In such cases, LLMs struggle to determine whether to rely more on their own parameters or the conflicted context. To address this, we propose **CK-PLUG**, a plug-and-play method for controlling LLMs' reliance on parametric and contextual knowledge. We introduce a novel knowledge consistency metric, *Confidence Gain*, which detects knowledge conflicts by measuring entropy shifts in token probability distributions after context insertion. CK-PLUG then enables fine-grained control over knowledge preference by adjusting the probability distribution of tokens with negative confidence gain through a single tuning parameter. Experiments demonstrate CK-PLUG's ability to significantly regulate knowledge reliance in counterfactual RAG scenarios while maintaining generation fluency and knowledge accuracy. For instance, on LLaMA3-8B, memory recall (MR) of RAG response can be adjusted within a broad range (9.9%-71.9%), compared to the baseline of 42.1%. Moreover, CK-PLUG supports adaptive control based on the model's confidence in both internal and external knowledge, achieving consistent performance improvements across various general RAG tasks. Our code is available at: `https://github.com/byronBBL/CK-PLUG`.

## 1 INTRODUCTION

Retrieval-Augmented Generation (RAG) (Lewis et al., 2020; Santhanam et al., 2021; Gao et al., 2023; Fan et al., 2024) has become a widely adopted technique for various applications, as it effectively integrates external knowledge with the powerful generative capabilities of Large Language Models (LLMs) (Achiam et al., 2023; Grattafiori et al., 2024) to produce accurate responses. However, potential knowledge conflicts (Xu et al., 2024a; Xie et al., 2023; Shi et al., 2025) between the external context and the model's internal parameters pose significant challenges to the reliability of RAG-generated outputs, often leading to hallucinations (Huang et al., 2023; Tonmoy et al., 2024).

There exists an inherent trade-off between the factuality of model parameters and the fidelity of externally retrieved context (Bi et al., 2024b). Enhancing the model's internal factuality (Chuang et al., 2023; Li et al., 2024a; Zhang et al., 2024b) may become unreliable as the model becomes outdated, while excessive dependence on retrieved context (Zhou et al., 2023; Shi et al., 2024) can be problematic due to the quality limitations of the retrieved information.

In this paper, we argue that efficient control of knowledge reliance is crucial for the effective deployment of RAG systems. Existing alignment to factuality (Tian et al., 2023; Lin et al., 2024a) or context faithfulness (Bi et al., 2024a; Huang et al., 2025a) are unidirectional and uncontrollable,

---

* Corresponding author.

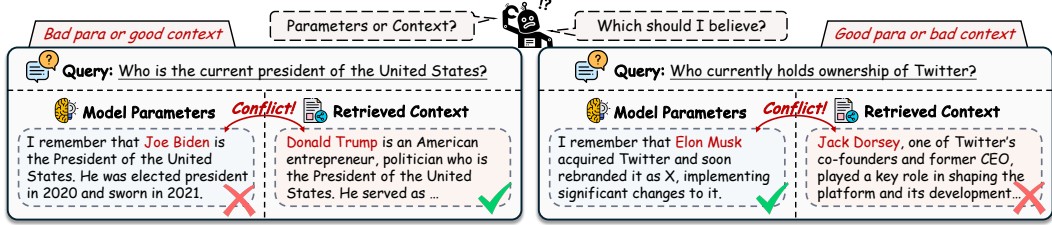

Figure 1: LLMs struggle to prioritize between parametric and contextual knowledge, especially when facing outdated parameters or misleading context, reducing reliability in real-world scenarios.

lacking the flexibility for bidirectional adjustment. The degree of reliance on internal parameters versus external context should be customizable to adapt to varying RAG scenarios, such as differences in model capabilities or retrieval quality. As illustrated in Figure 1, in the case of outdated models or high-quality or professional retrieval environments, the model should rely more on external knowledge. Conversely, when the retrieval context is noisy or potentially adversarial, the model should prioritize more its internal parameters to ensure reliable generation.

To achieve this, we propose CK-PLUG (**C**ontrollable **K**nowledge **Plug**-in), a pluggable inference-time approach for knowledge reliance control without modifying model parameters or architectures. To enable fine-grained adjustment, CK-PLUG introduces the *Confidence Gain* metric to detect knowledge conflicts. This metric quantifies the information gain of parameter-aware tokens after injecting contexts, measuring the consistency between parametric knowledge and external context.

Based on this metric, CK-PLUG retains tokens exhibiting positive confidence gains (indicating alignment between external context and the model's parametric knowledge) while dynamically adjusting the prediction strategy for tokens with negative confidence gains. For the latter, the framework blends parameter-aware and context-aware token probability distributions through a weighted fusion mechanism. The balance between these distributions is governed by a single tuning parameter $\alpha$, enabling fine-grained control over knowledge reliance preferences. Additionally, CK-PLUG introduces an automated mode that adaptively balances parametric and contextual reliance through entropy-based confidence evaluation, eliminating the need for manual $\alpha$ specification.

We evaluate CK-PLUG on various LLMs in RAG scenarios. Under explicit $\alpha$ control, the framework achieves substantial adjustments in Memory Recall (MR) for QA tasks with counterfactual retrieval contexts. For instance, on LLAMA3-8B, CK-PLUG modulates MR from 9.89% to 71.93%, significantly deviating from the baseline MR of 42.09%. In autonomous mode ($\alpha$-free), our CK-PLUG adaptively balances internal and external knowledge by leveraging model confidence metrics, yielding consistent performance gains across six distinct RAG downstream tasks. Our work paves the way for developing both knowledge-controllable and trustworthy generation capabilities for LLMs.

## 2 PRELIMINARY

**Language Model Generation** The current language model generation process aims to predict the next words within a given context sequence. Formally, given a sequence of tokens $X = \{x_1, x_2, \ldots, x_{t-1}\}$, LLMs process their embeddings $H = \{h_1, h_2, \ldots, h_{t-1}\}$ to compute the representation of the next token through transformer layers. An affine layer $\varphi(\cdot)$ is then applied to predict the next token distribution $x_t$ over the vocabulary set $\mathcal{V}$:

$$p(x_t|x_{<t}) = \text{softmax}(\phi(h_t)), \quad x_t \in \mathcal{V} \tag{1}$$

During decoding, various strategies can be applied to select the next token $x_t$ based on $p(x_t|x_{<t})$. This iterative process continues until the sequence generation reaches a designated end token or satisfies a predefined stopping condition. Our CK-PLUG controls knowledge reliance by adjusting the probability distribution of the next token during the decoding process.

**Perplexity Measured by Entropy** Entropy (Gray, 2011) is a fundamental concept in information theory that has been widely applied in natural language processing (NLP) (Pimentel et al., 2021; Vanmassenhove et al., 2021). It has proven particularly valuable in quantifying uncertainty within language modeling and generation tasks (Alon & Kamfonas, 2023; Meister et al., 2020). Given a

probability vector $\mathbf{a} \in \mathbb{R}^n$, where the entries are non-negative and the sum of all entries equals 1, the Shannon entropy is defined as follow:

$$H(\mathbf{a}) = -\sum_{i=1}^{n} a_i \log_2(a_i) \tag{2}$$

By quantifying the uncertainty in language model predictions, entropy can be used to measure the perplexity of LLMs. Building on this principle, we compute the entropy of the post-softmax probability distribution using Eq. (1) to measure the perplexity of next-token predictions in LLMs:

$$H(p(x_t|x_{<t})) = -\sum_{i=1}^{n} p_i \log_2(p_i) \tag{3}$$

Specifically, higher entropy values correspond to greater uncertainty in LLMs' next-token prediction, while lower entropy reflects deterministic confidence.

# 3  CK-PLUG: FINE-GRAINED KNOWLEDGE RELIANCE CONTROL

To address the challenge of dynamically balancing parametric and contextual knowledge in RAG systems, we propose CK-PLUG, a lightweight method that achieves granular control over language models' knowledge reliance via token-level probability modulation. In this section, we provide further details about our CK-PLUG. Section 3.1 introduces the knowledge conflict detection based on information gain, which serves as the operational switch for CK-PLUG. Section 3.2 explains the principle behind CK-PLUG's modulation of knowledge between parameters and context, while Section 3.3 discusses how CK-PLUG enables adaptive knowledge adjustment.

## 3.1  KNOWLEDGE CONFLICTS DETECTION WITH *Confidence-Gain*

CK-PLUG achieves fine-grained knowledge control through token-level probability modulation. Adjusting only key tokens can positively influence knowledge preference, whereas indiscriminately modifying all tokens can lead to a catastrophic collapse in generation quality (Lin et al., 2024b). To this end, we introduce a knowledge conflict detection mechanism as CK-PLUG's activation switch. This mechanism identifies tokens that exhibit potential conflicts between the LLM's parametric knowledge and the retrieved contextual knowledge, enabling targeted intervention.

First, we define the next-token prediction in model generation for a query $X_q$ as follows:

- $p(x|X_q)$ : Predictions conditioned solely on the input query $X_q$, reflecting the model's internal parametric knowledge.

- $p(x|X_r + X_q)$: Predictions conditioned on both query $X_q$ and retrieved context $X_r$, integrating parametric and external knowledge.

Here, the augmented distribution $p(x|X_r + X_q)$ serves as the objective of RAG precess, reflecting the LLM's response based on both its internal parameters and external context, while parametric distribution $p(x|X_q)$ represents predictions solely derived from the model's parametric knowledge.

Inspired by uncertainty quantification in token logits (Duan et al., 2023; 2024; Ma et al., 2025), we employ information entropy to measure prediction perplexity within generated distributions.. Based on Equation 3, we define $H(p(x|X_q))$ as the entropy of the parametric predictions and $H(p(x|X_r + X_q))$ as the entropy of the retrieval-augmented predictions.

We utilize the NQ dataset (Kwiatkowski et al., 2019) to evaluate the feasibility of entropy-based detection, along with *Conflict Contexts* (containing counterfacts contradicting parametric knowledge) and *Support Contexts* (retrieved factual evidence). We design a knowledge capture algorithm that aggregates the entropy of tokens corresponding to the decoded gold answer under both conflict and supportive conditions (see details in Appendix C.1 and D). For example, as depicted in Figure 3, when decoding terms such as "Dutch" or "Israel", we record the entropy of the token probability distributions at the relevant positions, which reflects the confidence in core knowledge.

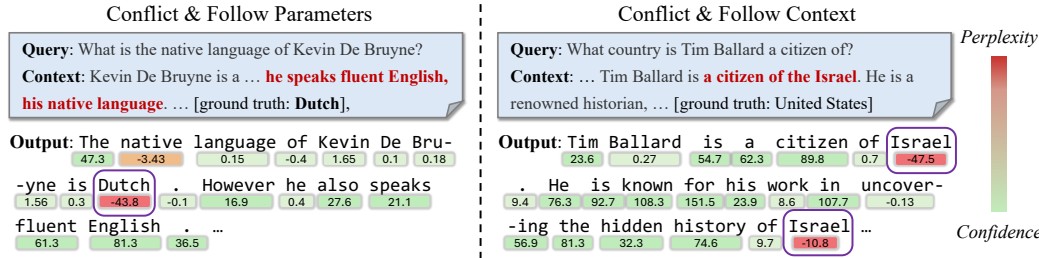

Figure 3: Illustration of the Confidence-Gain (*CG*) on LLaMA3-8B for generated tokens under two types of *Conflict Context*, demonstrating its effectiveness in detecting latent knowledge conflicts. For comparison, examples of *Support Context* are provided in the Appendix B.

Figure 2 compares the entropy changes before and after context insertion in both conflict and support scenarios. We observed that, in comparison, inserting *Conflict Context* increases entropy, reflecting a more disordered probability distribution and reduced confidence in model responses. In contrast, *Support Context* significantly decrease entropy, indicating that the model becomes more confident when its internal knowledge is corroborated by external information. Although the changes under conflict conditions are less pronounced in Figures 2 (c) and (d), the marked entropy reduction in supportive scenarios further highlights the model's confusion when faced with conflicting inputs.

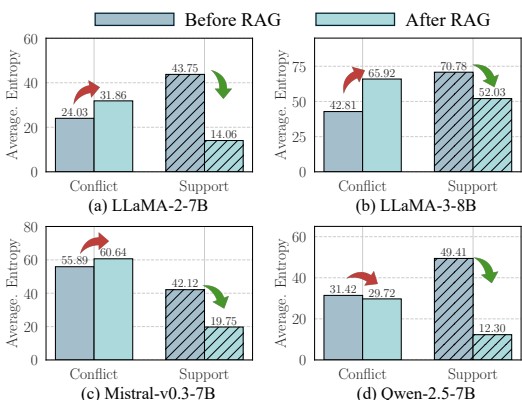

Figure 2: Changes (%) in the entropy of probability distribution for knowledge-sensitive tokens after incorporating conflict or support contexts.

Based on these observations, we propose a metric termed *Confidence Gain (CG)* to evaluate the change in model confidence before and after context insertion during decoding. Given the probability distributions $p(x|X_q)$ and $p(x|X_r + X_q)$, CG is computed as follows:

$$CG = H(p(x|X_q)) - H(p(x|X_r + X_q)) \qquad (4)$$

As shown in Figure 3, *CG* effectively measures the confidence shift of each token when incorporating retrieved context during generation. If the confidence drops significantly (i.e., *CG* falls below 0 or a predefined threshold specified in the Appendix B), the token is identified as a potential knowledge conflict. We then apply subsequent knowledge reliance modulation to these conflicting tokens.

## 3.2 PARAMETERS-CONTEXT RELIANCE MODULATION

In various RAG scenarios, the quality of retrieved texts may vary, necessitating user control over the reliance on either parametric knowledge or retrieved context. This control should be lightweight, avoiding the need to train multiple model versions. CK-PLUG efficiently achieves fine-grained knowledge reliance modulation by intervening in the probability distribution of the next-token prediction during the decoding phase.

During LLM inference in RAG, we define the parameter-aware log probability distribution as:

$$q_{\text{para}}(x|X_r + X_q) = \log p(x|X_q) \qquad (5)$$

where the query $X_q$ serves as the prompt, concatenated with the previously generated tokens in RAG as a prefill, to elicit the next-token prediction from the model's parametric knowledge. In contrast, the next-token prediction in RAG incorporates both parametric knowledge and retrieved context. By subtracting the parameter-aware log probability from the original log probability distribution, we

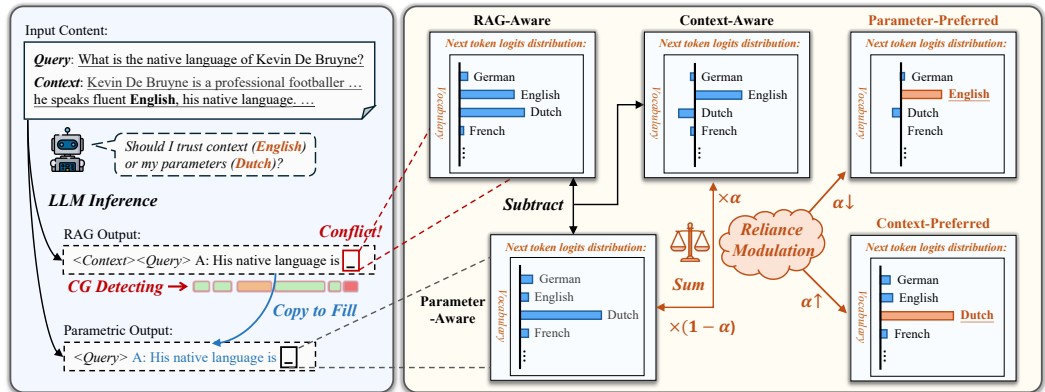

Figure 4: Illustration of CK-PLUG controlling the knowledge reliance in LLM outputs. During token generation, it detects potential conflicts and modulates the probability distribution of conflicted tokens. The modulation first computes a context-aware distribution, then integrates it with the parameter-aware distribution through a weighted sum based on the tuning parameter $\alpha$.

isolate the contribution of retrieved context, capturing its influence on token prediction. This leads to the definition of the context-aware distribution:

$$q_{\text{cont}}(x|X_r + X_q) = \log \frac{p(x|X_r + X_q)}{p(x|X_q)} \tag{6}$$

As shown in Figure 4, the core idea of CK-PLUG is to regulate knowledge reliance by modulating the parameter-aware and context-aware prediction distributions, particularly for tokens that indicate potential knowledge conflicts. Using $q(x)$ as a shorthand for $q(x|X_r + X_q)$, we compute the resulting distribution for next-word prediction as follows:

$$\hat{p}(x|X_r + X_q) = \begin{cases} \text{softmax}\Big(\mathcal{F}\big(q_{\text{cont}}(x), q_{\text{para}}(x)\big)\Big), & \text{if } CG < 0, \\ p(x|X_r + X_q), & \text{otherwise.} \end{cases} \tag{7}$$

where $CG$ represents the confidence gain metric, indicating whether retrieved context introduces conflicting information. We introduce a tunable hyperparameter $\alpha$ to control the balance between parametric and contextual reliance. The modulation function is defined as:

$$\mathcal{F}\big(q_{\text{cont}}(x), q_{\text{para}}(x)\big) = \begin{cases} \alpha \cdot q_{\text{para}} + (1 - \alpha) \cdot q_{\text{cont}}, & \text{if } x \in \mathcal{V}_{\text{head}}(x|X_r + X_q), \\ -\infty, & \text{otherwise.} \end{cases} \tag{8}$$

Following adaptive plausibility constraint (Li et al., 2022), we define the subset $\mathcal{V}_{\text{head}}(x|X_r + X_q) \subset \mathcal{V}$ as the union of the top-$k$ tokens from both parameter-aware and context-aware distributions:

$$\mathcal{V}_{\text{head}}(x|X_r + X_q) = \big\{x \in \mathcal{V} \mid q_{\text{para}}(x) > q_{\text{para}}\big(x_{\text{para}}^{R=k}\big)\big\} \cup \big\{x \in \mathcal{V} \mid q_{\text{cont}}(x) > q_{\text{cont}}\big(x_{\text{cont}}^{R=k}\big)\big\} \tag{9}$$

Here, $x^{R=k}$ represents the $k$-th ranked token in the parameter-aware or context-aware distribution. Taking their union ensures that context-related tokens with low confidence are also considered.

Through this modulation mechanism, we achieve controllable adjustment of the relative contributions of parametric and contextual knowledge. The reliance can be finely controlled with a single hyperparameter $\alpha$: increasing $\alpha$ makes the model more dependent on internal knowledge, while decreasing $\alpha$ shifts focus toward the retrieved context, even when it conflicts with parametric knowledge.

| Model | Method | NQ | | | ConFiQA | | | MQUAKE | | |
|-------|--------|------|------|------|------|------|------|------|------|------|
| | | ConR | ParR | MR | ConR | ParR | MR | ConR | ParR | MR |
| LLAMA2-7B-CHAT | Baseline | 43.3 | 43.8 | 50.2 (-) | 69.7 | 28.1 | 28.8 (-) | 31.2 | 21.6 | 40.9 (-) |
| | $\alpha = 0.0$ | 61.6 | 8.6 | **12.3** (↓75.5) | 71.5 | 9.2 | **11.4** (↓60.4) | 40.7 | 10.8 | **21.0** (↓48.7) |
| | $\alpha = 0.5$ | 45.6 | 32.2 | 41.4 (↓17.5) | 67.5 | 24.0 | 26.2 (↓9.0) | 24.6 | 14.6 | 41.7 (↑2.0) |
| | $\alpha = 1.0$ | 23.2 | 58.2 | **71.5** (↑42.4) | 31.5 | 46.2 | **59.4** (↑106.3) | 11.6 | 43.2 | **79.9** (↑95.4) |
| LLAMA3-8B-INSTRUCT | Baseline | 43.9 | 34.1 | 43.5 (-) | 54.2 | 22.4 | 29.2 (-) | 18.7 | 17.9 | 48.9 (-) |
| | $\alpha = 0.0$ | 63.5 | 7.3 | **9.9** (↓77.2) | 65.2 | 11.4 | **14.9** (↓48.9) | 42.1 | 15.5 | **26.9** (↓45.0) |
| | $\alpha = 0.5$ | 44.7 | 32.5 | 42.1 (↓3.2) | 51.7 | 17.9 | 25.7 (↓12.0) | 20.2 | 20.4 | 50.3 (↑2.9) |
| | $\alpha = 1.0$ | 22.5 | 57.6 | **71.9** (↑65.3) | 25.4 | 42.3 | **62.5** (↑114.0) | 14.5 | 47.1 | **76.5** (↑56.4) |
| MISTRAL0.3 7B-INSTRUCT | Baseline | 46.2 | 58.6 | 55.9 (-) | 64.7 | 25.9 | 28.6 (-) | 43.8 | 21.2 | 32.6 (-) |
| | $\alpha = 0.0$ | 75.8 | 15.8 | **17.2** (↓69.3) | 70.7 | 10.9 | **13.4** (↓53.1) | 65.8 | 12.4 | **15.9** (↓51.2) |
| | $\alpha = 0.5$ | 46.2 | 58.6 | 55.9 (-) | 65.7 | 25.9 | 28.6 (-) | 43.5 | 22.2 | 33.8 (↑3.7) |
| | $\alpha = 1.0$ | 27.9 | 69.1 | **72.2** (↑29.2) | 29.9 | 43.8 | **59.5** (↑108.0) | 15.2 | 50.4 | **76.8** (↑135.6) |
| QWEN2.5-7B-INSTRUCT | Baseline | 73.4 | 32.4 | 31.3 (-) | 43.8 | 15.4 | 26.1 (-) | 32.2 | 13.0 | 28.8 (-) |
| | $\alpha = 0.0$ | 85.4 | 8.3 | **9.0** (↓71.3) | 65.2 | 13.9 | **17.6** (↓32.5) | 49.3 | 12.8 | **20.6** (↓28.5) |
| | $\alpha = 0.5$ | 72.0 | 26.8 | 27.1 (↓13.4) | 43.8 | 14.9 | 25.4 (↓2.7) | 32.8 | 13.2 | 28.8 (-) |
| | $\alpha = 1.0$ | 30.2 | 51.4 | **63.2** (↑101.9) | 36.8 | 28.4 | **43.5** (↑66.7) | 19.8 | 32.8 | **62.4** (↑116.7) |

Table 1: Performance (%) of CK-PLUG in controlling knowledge reliance, with $\alpha$ set to 0.0, 0.5, and 1.0. **Red** markers denote sharp MR decreases indicating enhanced contextual alignment, while **green** markers highlight significant MR increases reflecting strengthened parametric reliance.

### 3.3 ADAPTIVE KNOWLEDGE ADJUSTMENT

CK-PLUG also can autonomously balances parametric and contextual dependencies through entropy-based perplexity. For notational brevity, let $H_{\text{para}}$ replace $H(p(x|X_q))$ to represent parametric perplexity and $H_{\text{cont}}$ replace $H(p(x|X_r + X_q))$ to denote contextual perplexity after retrieval injection. Since higher entropy corresponds to lower model confidence, we reformulate the modulation parameter $\alpha$ in Equation 8 as a normalized ratio of perplexities:

$$\alpha = \frac{H_{\text{cont}}}{H_{\text{para}} + H_{\text{cont}}} \tag{10}$$

This eliminates manual $\alpha$-specification, enabling CK-PLUG to explicitly balance knowledge reliance based on the model confidence, enhancing both interpretability and trustworthiness in generation.

## 4 EXPERIMENTAL METHODOLOGY

**Models and Tasks** We integrate CK-PLUG into the generation process of LLMs by modifying the decoding operation. Our experiments evaluate the performance of CK-PLUG on four popular open-source LLMs: LLAMA2-7B (Touvron et al., 2023), LLAMA3-8B (Grattafiori et al., 2024), MISTRALV0.3-7B (Jiang et al., 2023a), and QWEN2.5-7B (Yang et al., 2024). We assess CK-PLUG's effectiveness in both knowledge reliance control (Section 3.2) and adaptive generation enhancement (Section 3.3). See Appendix C.1 for details about the datasets and implementation.

**Evaluation for Knowledge Control** To evaluate the effectiveness of CK-PLUG in modulating the reliance on parametric and contextual knowledge, we simulate a RAG environment with knowledge conflicts. Specifically, we modify the retrieved contexts in the NQ dataset to contain factually incorrect statements related to the answers, following Longpre et al. (2021). Additionally, we incorporate ConFiQA (Bi et al., 2024a) and MQuAKE (Zhong et al., 2023), which provide noisy counterfactual contexts and knowledge editing instructions, respectively. These tasks introduce counterfactual information that conflicts with the model's parametric knowledge. We use ConR

| Model | Method | NQ | HotpotQA | FEVER | T-REX | Eli5 | WOW | Avg |
|---|---|---|---|---|---|---|---|---|
| LLaMA2-7B-chat | w/o RAG | 30.1 | 27.5 | 55.9 | 11.8 | 13.8 | 13.6 | 25.5 |
| | w/ RAG | 41.4 | 40.9 | 66.2 | **42.3** | 13.3 | 14.5 | 36.4 |
| | RAG w/ CK-PLUG | **43.7** | **42.6** | **72.6** | 41.7 | **14.1** | **15.2** | **38.3** |
| LLaMA3-8B-instuct | w/o RAG | 32.1 | 32.2 | 73.6 | 19.1 | 14.0 | 13.1 | 30.7 |
| | w/ RAG | 45.2 | 43.1 | **86.4** | 51.5 | 14.0 | 13.6 | 42.3 |
| | RAG w/ CK-PLUG | **46.5** | **46.7** | 86.1 | **52.3** | **14.8** | **14.3** | **43.5** |
| Mistral0.3-7B-instruct | w/o RAG | 34.7 | 32.3 | 74.2 | 29.3 | 15.7 | 13.9 | 33.4 |
| | w/ RAG | 47.8 | 44.9 | **89.5** | 57.8 | 15.5 | 14.8 | 45.1 |
| | RAG w/ CK-PLUG | **49.5** | **46.2** | 89.2 | **58.1** | **15.8** | **15.3** | **45.7** |
| Qwen2.5-7B-instruct | w/o RAG | 27.7 | 27.1 | 56.4 | 25.0 | 14.2 | 13.0 | 27.2 |
| | w/ RAG | 49.5 | 47.9 | 85.6 | **61.6** | 13.8 | 14.0 | 45.4 |
| | RAG w/ CK-PLUG | **50.3** | **48.5** | **87.8** | 60.2 | **14.3** | **14.5** | **45.9** |

Table 2: Results (%) on the adaptive enhancement of CK-PLUG across six diverse RAG tasks.

(the recall of context) and ParR (the recall of parameters). ConR measures whether the generated responses align with the provided context, while ParR evaluates their alignment with the model's parametric knowledge. Specifically, we also adopt the memorization ratio $MR = \frac{ParR}{ParR+ConR}$, which captures the tendency to favor parametric knowledge over retrieved context.

**Evaluation for Adaptive Enhancement**   We evaluate the effectiveness of CK-PLUG's adaptive adjustment in a general RAG setting. Specifically, we use Wikipedia[1] as the corpus and BGE-base-v1.5 (Xiao et al., 2023) as the retriever. Our evaluation covers six diverse RAG tasks from the KILT benchmark (Petroni et al., 2021), including *Open-Domain QA* on NQ (Kwiatkowski et al., 2019), *Multi-Hop QA* on HotpotQA (Yang et al., 2018), *Fact Verification* on FEVER (Thorne et al., 2018), *Slot Filling* on T-REX (Elsahar et al., 2018), *Long-Form QA* on ELI5 (Fan et al., 2019), and *Dialogue Generation* on WOW (Dinan et al., 2019). Specifically, we use normalized accuracy to evaluate the first four tasks, while Rouge-L and F1 scores are used to assess ELI5 and WOW, respectively.

## 5 Evaluation Results

### 5.1 Overall Performance

**CK-PLUG Enables Wide-Range Knowledge Reliance Control**   Table 1 presents the knowledge control results of CK-PLUG across different evaluation settings. Specifically, NQ evaluates standard QA, ConFIQA assesses long-context QA, and MQuAKE examines multi-turn QA, all designed to measure knowledge reliance under counterfactual contexts. Compared to the baseline, when $\alpha = 0.0$, CK-PLUG enhances context reliance (increased ConR) while reducing reliance on parametric knowledge (decreased ParaR). Conversely, at $\alpha = 1.0$, the trend is reversed. The substantial variation in MR further underscores CK-PLUG's effectiveness in controlling knowledge reliance. For instance, on LLaMA2-7B, CK-PLUG adjusts MR over a broad range, from 14.9% to 70.3% on average. Furthermore, at $\alpha = 0.5$, the model's performance closely aligns with the baseline, exhibiting only minor fluctuations. This suggests that CK-PLUG effectively balances parametric and contextual knowledge alignment with the model's inherent knowledge attention, aligning with our expectation of smooth and linear modulation of knowledge preference.

**CK-PLUG Enhances Generation Reliability with Adaptive Control**   CK-PLUG autonomously adjusts $\alpha$ to enhance generation reliability. As shown in Table 2, CK-PLUG improves overall performance across six distinct tasks compared to baselines with or without retrieved contexts. These results demonstrate CK-PLUG's ability to strengthen reliability through adaptive parametric-contextual knowledge balancing. Notably, when the performance of systems without RAG and with RAG are close, which suggests that parameter and contextual knowledge contribute differently to reliable generation, CK-PLUG effectively balances them to achieve a more significant improvement.

### 5.2 Fine-Grained Control via a Single Tuning Parameter

---

[1]Dump from http://dl.fbaipublicfiles.com/BLINK/enwiki-pages-articles.xml.bz2

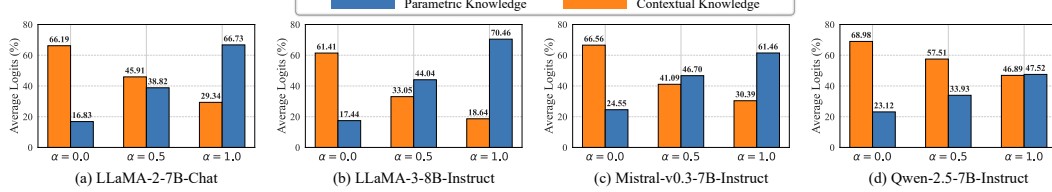

Figure 6: Average probabilities (%) of the parametric and contextual knowledge components in knowledge-aware tokens, which increase and decrease respectively with increasing parameter $\alpha$.

CK-PLUG employs a single parameter $\alpha$ to regulate the model's reliance on contextual knowledge versus parameterized knowledge. Figure 5 illustrates the impact of fine-grained adjustments on MR within the NQ dataset, which includes counterfactual contexts. Due to intrinsic differences among models, the variation in MR as $\alpha$ changes exhibits slight discrepancies. Notably, while most models follow a highly consistent pattern, QWEN2.5-7B shows a distinct behavior, particularly when $\alpha \geq 0.5$, where the increase in MR slows down. This observation aligns with prior findings by Bi et al. (2024a), which suggest that QWEN models tend to be more confident in its parametric knowledge when it conflicts with the provided context. Nevertheless, the trend remains approximately linear, ensuring smooth modulation and CK-PLUG's adaptability across applications.

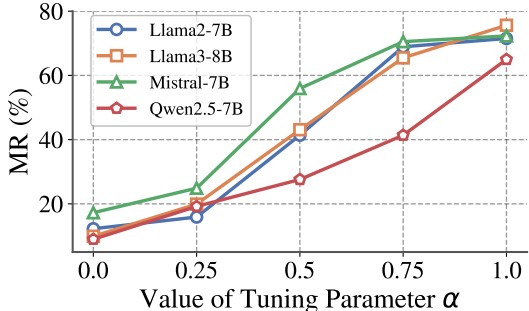

Figure 5: Variation in MR (%) across different language models as parameter $\alpha$ increases.

## 5.3 ABLATION STUDY

Knowledge conflict detection (ConD) is a crucial component of CK-PLUG, ensuring that knowledge modulation is applied only to tokens that could potentially trigger conflicts (Section 3.1). Without this selective adjustment, excessive modulation may lead to catastrophic generation failures. To validate the importance of this module, we conduct an ablation study on the NQ dataset with counterfactual contexts. Specifically, we use the hit rate as a metric to evaluate generation quality, measuring whether the model output contains either the original parametric answer or the gold answer from the context. The results, presented in Table 3, show that CK-PLUG with ConD maintains a hit rate comparable to the baseline

| Model | Setting | Baseline | a = 0.0 | a = 0.5 | a = 1.0 |
|---|---|---|---|---|---|
| LLAMA2-7B-CHAT | Baseline | 78.9 | - | - | - |
| | w/ ConD | - | 74.8 | 77.8 | 78.5 |
| | w/o ConD | - | **30.7** | **58.4** | **62.3** |
| LLAMA3-8B-INSTRUCT | Baseline | 83.7 | - | - | - |
| | w/ ConD | - | 82.4 | 83.5 | **86.7** |
| | w/o ConD | - | **53.8** | **61.4** | **73.2** |
| MISTRAL-INSTRUCT | Baseline | 92.4 | - | - | - |
| | w/ ConD | - | 89.5 | **92.4** | 89.7 |
| | w/o ConD | - | **49.9** | 91.3 | **75.4** |
| QWEN-INSTRUCT | Baseline | 88.8 | - | - | - |
| | w/ ConD | - | 89.6 | **89.8** | 87.2 |
| | w/o ConD | - | **62.4** | 85.4 | **51.3** |

Table 3: Hit rate (%) of our CK-PLUG with and without conflict detection (ConD). The *Baseline* represents standard RAG without CK-PLUG.

across different models. In contrast, removing ConD leads to a noticeable decline (**highlighted**), particularly in LLAMA models and under extreme knowledge modulation settings ($\alpha$=0.0 or $\alpha$=1.0). This demonstrates that ConD effectively identifies tokens requiring modulation, ensuring reliable generation while preventing the risks associated with excessive adjustments.

## 5.4 DEEP INSIGHTS INTO KNOWLEDGE BETWEEN PARAMETERS AND CONTEXT

The previous results have demonstrated CK-PLUG's effectiveness in controlling knowledge reliance. However, a deeper analysis is necessary to ensure the reliability of this modulation. In this section, we investigate the impact of CK-PLUG on model outputs from an interpretability perspective.

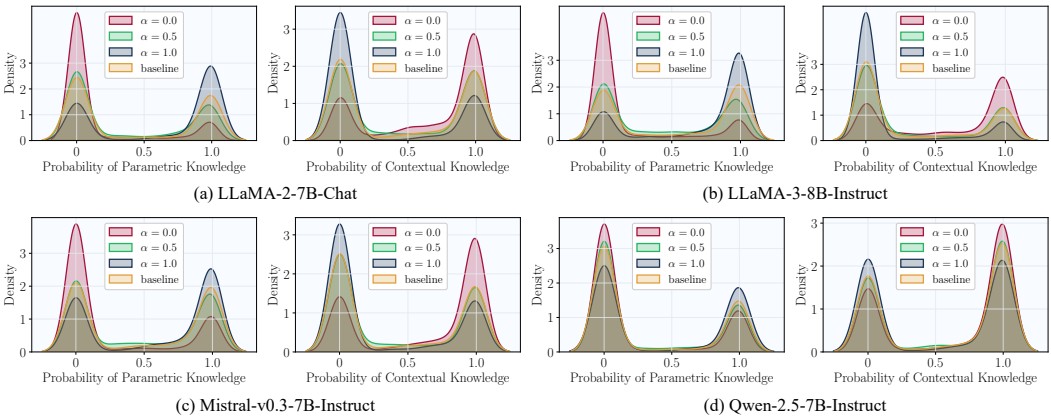

Figure 7: Experimental results of kernel density estimation for the softmax probability distribution of tokens reflecting parametric and contextual reliance across different $\alpha$ settings.

To achieve this, we design a specialized algorithm to capture the probability distribution of the first token in the model's response that reflects knowledge reliance. For example, given the query, "*In which country is London located?*" with the provided context, "*London is a city in France*", a parametric response might be "*London is located in England*" while a context-dependent response would be "*London is located in France*". The algorithm automatically detects the first decoded token corresponding to "England" or "France" (or their prefix substrings like "Eng-" or "Fran-"), effectively capturing the model's knowledge reliance. Based on this, we obtain the probability of this token being generated under parametric or contextual dependence (e.g., "English" and "Dutch" in Figure 3). The details of the algorithm is provided in the Appendix D.

We apply this token-level probability analysis to the ConFIQA dataset. Figure 6 presents the average probabilities of parametric and contextual knowledge under different values of $\alpha$. As $\alpha$ increases, the probability of contextual knowledge decreases, while the probability of parametric knowledge correspondingly increases. This aligns with our previous observations in Section 5.2, where QWEN2.5-7B exhibits strong confidence in its parametric knowledge when unreliable context is introduced. Figure 7 provides a more detailed probability distribution analysis: for parametric knowledge, smaller $\alpha$ values concentrate probabilities in the lower range, while larger $\alpha$ values shift them to the higher range; for contextual knowledge, the trend is reversed. These fine-grained results offer deeper insights into CK-PLUG's behavior, illustrating how it effectively modulates knowledge dependence at the token level to control the model's knowledge preference in generation.

## 5.5 CASE STUDY

Table 4 presents case studies on NQ dataset, showing that CK-PLUG significantly alters LLM outputs while maintaining fluency and logical consistency. This suggests CK-PLUG fundamentally regulates knowledge reliance rather than merely forcing token-level changes. More cases are in Appendix E.

## 6 CONCLUSION

In this work, we argue that LLMs should have personalized knowledge reliance preferences tailored to different RAG scenarios. We introduce CK-PLUG, a plug-and-play method for controlling LLMs' reliance on parametric and contextual knowledge. We use *Confidence Gain* to detect potential conflicts in generated tokens and apply a single parameter to modulate the token probability distribution between parametric and contextual components for tokens with negative confidence gain. Additionally, CK-PLUG offers an adaptive method for adjusting knowledge reliance to enhance generation reliability. Experimental results demonstrate that CK-PLUG enables smooth control over knowledge reliance while maintaining generation coherence, and consistently improves performance across a wide range of RAG tasks. Our findings emphasize the need for explicit knowledge reliance control and offer a practical framework for balancing parametric and contextual knowledge in LLMs.

| | | |
|---|---|---|
| **LLAMA3-8B** | *Query* | Where was the capital of the habsburg empire located? [Ground Truth: Vienna] |
| | *Context* | The Habsburg Monarchy (German: Habsburgermonarchie) or Empire is an unofficial appellation among historians for the countries and provinces... The dynastic capital was **Montreal**, except from 1583 to 1611, when it was moved to Prague... |
| | *Output* | The Dynastyczal capital **Montreal** except from 1583 to 1611 when it was moved to Pragma. From 1804 to 1867 the Haburg Monarchy was formely unified as... **Context** *Reliance* ($\alpha$=0.0) |
| | | The capital of the Habsburg Empire was **Vienna**, Austria. **Parameter** *Reliance* ($\alpha$=1.0) |
| **MISTRAL-7B** | *Query* | When did the us not go to the olympics? [Ground Truth: 1980] |
| | *Context* | The **1981** Summer Olympics boycott was one part of a number of actions initiated by the United States to protest the Soviet invasion of Afghanistan. The Soviet Union, which hosted the 1981 Summer Olympics, and other countries would later boycott the 1984 Summer Olympics in Los Angeles. |
| | *Output* | In **1981**, the United States led a boycott of the Summer Olympics in Moscow in protest of the Soviet invasion of Afghanistan. The following year, the Soviet Union and other countries... **Context** *Reliance* ($\alpha$=0.0) |
| | | The United States has participated in every modern Olympic Games since the 1896 Summer Olympics in Athens, Greece, except for the **1980** Summer Olympics in Moscow... **Parameter** *Reliance* ($\alpha$=1.0) |

Table 4: Case studies on LLAMA3-8B and MISTRALV0.3-7B for CK-PLUG's knowledge control ($\alpha = 0.0$ and $\alpha = 1.0$). **Green** text indicates the ground truth and its parametric match in the output while **red** denotes the counterfactual content in context and its corresponding faithful match.

## ACKNOWLEDGMENTS

This work is supported in part by the National Key R&D Program of China under Grant Nos. 2023YFA1011602, Beijing Natural Science Foundation No. 4262033, and the National Natural Science Foundation of China under Grant Nos. U25B2076, 62441229, and 62377043.

## ETHICS STATEMENT

This work does not involve human subjects, personally identifiable information, or sensitive user data. All experiments are conducted on publicly available datasets (NQ, HotpotQA, FEVER, T-REX, ELI5, and WoW) and widely used open-source models (e.g., LLaMA, Mistral, Qwen). We follow the licenses and terms of use of all datasets and models. Our study focuses on improving the reliability of retrieval-augmented generation and does not aim to promote harmful or discriminatory applications. The methods and findings are intended solely for research purposes, and care should be taken to avoid potential misuse in contexts where misinformation could have negative social impact.

## REPRODUCIBILITY STATEMENT

We have taken multiple steps to ensure reproducibility. Details of the experimental setup, datasets, and evaluation metrics are provided in the main paper (Sections 4–5) and Appendix C. Dataset preprocessing steps and task prompts are included in the supplementary materials. Hyperparameters and implementation details of CK-PLUG are reported in Appendix C. In addition, we provide an anonymous code repository with full implementations and experiment scripts at https://anonymous.4open.science/r/CK-PLUG-Ano-8E62. These materials enable other researchers to replicate our results and extend our work.

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

## A    RELATED WORK

Hallucinations in large language models (LLMs) have drawn significant research attention due to their adverse effects on generating unreliable or factually inconsistent content (Tonmoy et al., 2024; Huang et al., 2023; Wang et al., 2023; Mei et al., 2024). These issues are particularly critical in high-stakes domains where factual accuracy is paramount, prompting extensive efforts to detect and mitigate hallucinations (Gunjal et al., 2024; Liu et al., 2024; Zhang et al., 2024a; Ni et al., 2025). Various tools (Nakano et al., 2022; Yao et al., 2023; Qin et al., 2024) and retrieval-augmented generation (RAG) methods (Guu et al., 2020; Izacard & Grave, 2021; Huang et al., 2025b) have emerged as promising solutions by grounding model outputs in external knowledge sources. However, unresolved challenges persist in managing knowledge conflicts—the discrepancies between retrieved evidence and the model's internal knowledge. These conflicts manifest in three primary forms: (1) intra-parameter conflicts (inconsistencies within model parameters), (2) inter-context conflicts (contradictions across retrieved passages), and (3) parameter-context conflicts (mismatches between parametric knowledge and retrieved evidence). The latter poses a critical bottleneck for reliable RAG deployment, as it directly undermines trustworthiness in dynamically evolving knowledge scenarios.

Existing approaches predominantly address intra-parameter and inter-context conflicts through hallucination mitigation techniques (Luo et al., 2024; Zhang et al., 2023; Xu et al., 2024b; Li et al., 2024a) or retrieval augmented strategies (Ram et al., 2023; Jiang et al., 2023b; Fan et al., 2024). Parameter-context conflicts, however, remain undertheorized due to their inherent opacity: The interplay between a model's parametric knowledge and contextual evidence operates as a "dark mechanism" with limited interpretability. Recent attempts (Wang et al., 2024; Li et al., 2024b; Wei et al., 2024) to resolve this issue employ auxiliary models or agent-based systems to arbitrate knowledge reliability, yet these methods lack both explainability and adaptability to human preferences in real-time generation. Parallel efforts focus on unilateral enhancements—either refining parametric factuality through model editing (Meng et al., 2022; Fang et al., 2024; Li et al., 2025; Zhang et al., 2025) or improving contextual faithfulness (Zhou et al., 2023; Huang et al., 2025a). Such approaches, while effective in specific cases, prove inadequate for diverse RAG scenarios requiring flexible knowledge reliance control.

This work introduces a plug-and-play control framework that dynamically adjusts knowledge reliance preferences during generation. Unlike prior methods constrained by static architectures or targets, CK-PLUG enables scenario-specific adaptation through human-aligned mechanisms, addressing the fundamental limitations of existing conflict resolution paradigms in RAG systems.

## B    DETALIS OF CONFIDENCE GAIN

Conflict detection is based on the *Confidence Gain* computed during token generation, as demonstrated in Figure 3 under conflict contexts. Figure 8 presents the *Confidence Gain* distribution in supportive contexts. Here, the supportive context reinforces the model's parametric knowledge (e.g., "entrepreneurship"), which is reflected in the high *Confidence Gain* values for the corresponding token. This indicates that detecting the information gain after context insertion effectively assesses the consistency between contextual and parametric knowledge.

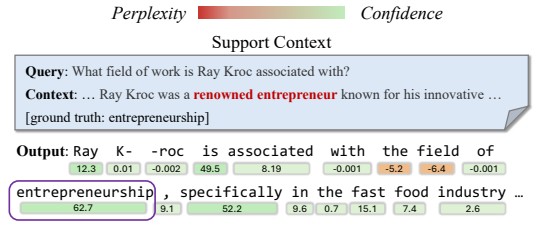

Figure 8: Example of the Confidence-Gain (CG) on LLAMA3-8B for generated tokens under *Support Context*.

Although the *Confidence Gain* metric effectively identifies conflict situations, slight differences in the probability mapping of internal knowledge across models necessitate a more precise conflict detection. Therefore, we extend the CG condition in Equation 8 for more accurate discrimination:

$$\hat{p}(x|X_r + X_q) = \begin{cases} \mathrm{softmax}\big(\mathcal{F}\big(q_{\mathrm{cont}}(x), q_{\mathrm{para}}(x)\big)\big), & \text{if } CG < \varepsilon \cdot |H(p(x|X_r + X_q))|, \\ p(x|X_r + X_q), & \text{otherwise.} \end{cases} \tag{11}$$

The above equation allows for finer-grained control over detection sensitivity across different models. In our experiments, we set the detection threshold $\varepsilon$=-2,-1,-1,-3 for LLAMA2-7B, LLAMA3-8B, MISTRALV0.3-7B and QWEN2.5-7B, respectively, ensuring a stricter token filtering mechanism to prevent excessive modifications.

## C EXPERIMENTAL SETUP

### C.1 DATASETS

#### C.1.1 DATA FOR KNOWLEDGE CONTROL

To evaluate CK-PLUG's ability to effectively control the model's knowledge dependency, we inject factually incorrect but query-relevant information into the retrieved context during the RAG process. We then observe whether the model's output aligns with the injected false context or adheres to the ground truth encoded in its parameters. The datasets used in our evaluation are as follows:

- **NQ** is a widely used question-answering dataset constructed with Wikipedia. Following Longpre et al. (2021), we replace the gold entity answer in the context (retrieved from corpus according to each question) with a randomly sampled entity of the same type from the corpus, thereby modifying the context to support a counterfactual answer. NQ with counterfactual context data can be found at [2].

- **ConFiQA** (Bi et al., 2024a) is a novel dataset designed to assess context-faithfulness in question-answering tasks using counterfactual retrieval passages. It evaluates whether models can generate responses that align with contexts containing counterfactual elements, simulating real-world scenarios where knowledge conflicts arise in modern RAG systems. For our evaluation, we specifically use the ConFiQA-QA subset to assess RAG performance under counterfactual contexts. The dataset can be found at [3].

- **MQuAKE** (Zhong et al., 2023) introduces multi-hop knowledge questions embedded with extensively modified facts, serving as a crucial benchmark for assessing knowledge editing in counterfactual settings. Unlike the previously mentioned datasets, MQuAKE not only features multi-hop QA but also integrates instructional counterfactual contexts, enabling a more rigorous evaluation of a model's reliance on encoded knowledge. The dataset (MQuAKE-CF-3k-v2.json) is available at [4].

#### C.1.2 DATA FOR ADAPTIVE ENHANCEMENT

For the evaluation of adaptive enhancement, we select six datasets covering various knowledge-intensive RAG tasks from KILT (Petroni et al., 2021). Below, we provide a detailed description of each dataset:

- **NQ** (Kwiatkowski et al., 2019) is a widely used open-domain question-answering dataset based on Wikipedia. The questions are sourced from Google search queries, and the answers are extracted as text spans from relevant Wikipedia articles. There are 3.6k questions in total.

- **HotpotQA** (Yang et al., 2018) is a multi-hop question-answering dataset that requires reasoning across multiple passages to derive the correct answer. The dataset includes both supporting facts and answers, facilitating research on multi-document retrieval and reasoning.

- **FEVER** (Thorne et al., 2018) is a fact verification dataset designed for verifying factual claims against Wikipedia evidence. Each claim is labeled as either "Supports" or "Refutes" based on retrieved supporting passages, making it a benchmark for automated fact-checking systems.

- **T-REX** (Elsahar et al., 2018) is a slot-filling dataset that focuses on knowledge base completion. Given an entity and a relation, the model must predict the missing object in the triple. The dataset is derived from Wikidata and aligned with textual mentions in Wikipedia, enabling studies on knowledge representation and extraction.

---

[2] https://drive.google.com/file/d/1DJ1ajmLNAKVTBWnM7SkP93EYQ2cav3Mk/view
[3] https://github.com/byronBBL/Context-DPO/tree/master/ConFiQA
[4] https://github.com/princeton-nlp/MQuAKE/tree/main/datasets

- **ELI5** (Fan et al., 2019) is a long-form question-answering dataset that contains open-ended questions from the "Explain Like I'm Five" subreddit. The dataset emphasizes generating detailed, explanatory, and well-structured answers, making it suitable for research in abstractive summarization and complex answer generation.
- **WOW** (Dinan et al., 2019) is a dialogue generation dataset in which agents generate informative and engaging responses based on Wikipedia passages. It is designed for knowledge-grounded conversation and requires models to integrate retrieved knowledge into responses naturally.

For each dataset, we randomly sample 1,000 data to serve for our evaluation of general RAG tasks. For the external corpus, we employ the Wikipedia, specifically the dump dated 2019-08-01. Following Wang et al. (2019), we conduct segmentation by splitting the original articles into segments with a maximum length of 100 words, which finally results in a total of 28,773,800 passages. For the retriever in our experiment, we utilize the BGE-base-en-v1.5 (Xiao et al., 2023), which shows a competitive performance on retrieval benchmarks, such as MTEB (Muennighoff et al., 2022). This model has 109M parameters and an embedding dimension of 768. We employ the cosine similarity to calculate the ranking score for each pair of query embedding and passage embedding.

## C.2 METRICS

**Konwledge Control** To assess knowledge reliance, we introduce ConR (context recall) and ParR (parameter recall). ConR quantifies the extent to which generated responses adhere to the retrieved context, whereas ParR reflects their consistency with the model's intrinsic knowledge. Additionally, we define the memorization ratio as $MR = \frac{ParR}{ParR + ConR}$, which indicates the degree to which the model prioritizes its parametric knowledge over external information.

**Adaptive Enhancement** To evaluate the overall enhancement of RAG tasks through adaptive knowledge adjustment, we first normalize both the gold answers and the model's outputs. Accuracy is used to assess performance on four tasks: open-domain QA on NQ, multi-hop QA on HotpotQA, fact verification on FEVER, and slot filling on T-REX. Meanwhile, Rouge-L and F1 scores are employed to evaluate long-form QA on ELI5 and dialogue generation on WOW.

Additionally, in Section 5.3, we employ hit rate to assess the fluency and logical consistency in model generation, evaluating whether its responses adhere to the counterfactual answers from the retrieved context or the ground truth encoded in its parameters. The specific task prompts for each task can be found in Appendix C.3.

## C.3 IMPLEMENTATION DETAILS

### C.3.1 KNOWLEDGE CONTROL

We use the following prompt template to obtain the model's output based on the input question and either the counterfactual context or the provided instructions.

**NQ/ConFiQA/MQuAKE:**

```
Background: {couterfactual context/instruction}

Q: {Input Query}

A: {LLM Output}
```

### C.3.2 ADAPTIVE ENHANCEMENT

We set the model output parameters to max_token=64 and top_k=100, using the top 10 retrieved contexts from BGE. We use the following prompt to conduct standard RAG experiments.

**NQ/HotpotQA/ELI5/T-REX/FEVER/WOW:**

```
Background:
```

```
Passage 1: {Retrieved Top Passage 1}

Passage 2: {Retrieved Top Passage 2}

Passage 3: {Retrieved Top Passage 3}

...

{Task Instruction}

Q: {Input}

A: {LLM Output}
```

The task instructions are presented in Table 5.

| Dataset | Task | Task Instruction | Example Data |
|---------|------|------------------|--------------|
| NQ | Open-domain QA | Answer the question based on the given passages. | Q: Who had the most wins in the nfl? 
 A: Tom Brady |
| HotpotQA | Multi-hop QA | Answer the question based on the given passages. You may need to refer to multiple passages. | Q: Which American politician did Donahue replaced? 
 A: Kelli Ward |
| ELI5 | Long-form QA | Answer the question based on the given passages. The answer needs to be detailed, paragraph-level, and with explanations. | Q: Why are the things that taste the best bad for us? 
 A: Let's think about this from an evolutionary perspective.  Way back in the day... |
| FEVER | Fact Verification | Verify whether the claim is correct based on the given passages.  If it is correct, output "SUPPORTS", if it is wrong, output "REFUTES". | Q: There is a movie called The Hunger Games. 
 A: SUPPORTS |
| WoW | Dialogue Generation | Generate an appropriate, reasonable and meaningful response based on previous conversations and the following relevant passages. | Q: Ever heard of Yves Saint Laurent?\nNope, what/who are they.\nThey are a French luxury fashion house.\nOh really who founded it? 
 A: Yep!  It was founded by Yves Saint Laurent, believe it or not. |
| T-REx | Slot Filling | Given an entity and an attribute (or relationship), fill in the specific value of the attribute based on the following passages.  The entity and the attribute are separated by "[SEP]". | Q: Serge Blisko [SEP] occupation 
 A: politician |

Table 5: Task instruction and example data of each dataset.

## D  KNOWLEDGE CAPTURE FOR CRUCIAL TOKENS

In this paper, we control model outputs by modulating the token probability distribution in the presence of potential knowledge conflicts.  To demonstrate the interpretability of entropy-based

---

**Algorithm 1** Knowledge Token Capturing

---

**Require:** The LLM generates a token sequence of length $n$, $\mathcal{V}$: vocabulary of the LLM, $\mathcal{P}_i \in (\mathcal{P}_1, \mathcal{P}_2, \ldots, \mathcal{P}_n)$: the logits distribution for each token, $S_{\text{cont}}$: string of the contextual answer (from the counterfactual context), $S_{\text{para}}$: string of the parametric answer (from the ground truth).
**Ensure:** Captured contextual knowledge logits $P_{\text{cont}}$ and parametric knowledge logits $P_{\text{para}}$.

1: Initialize $P_{\text{cont}} \leftarrow$ *None*, $P_{\text{para}} \leftarrow$ *None*
2: $S_{\text{com}} \leftarrow \text{COM}(S_{\text{cont}}, S_{\text{para}})$               ▷ Identify common substrings
3: **for** $\mathcal{P}_i \in (\mathcal{P}_1, \mathcal{P}_2, \ldots, \mathcal{P}_n)$ **do**
4:      Let $x_i \leftarrow \arg\max \mathcal{P}_i$ and $x_i' \leftarrow \text{Decode}(x_i)$.     ▷ Greedy decodes the location token
5:      **if** $x_i' \notin S_{\text{cont}}$ **and** $x_i' \notin S_{\text{para}}$ **then**
6:          **continue**         ▷ Skip if the highest probability token is not in either answer.
7:      **end if**
8:      **for** each token $x_j \in \mathcal{V}$ **(sorted in descending order by $P_{i,j}$) do**
9:          Decode $x_j$ into string $x_j'$.
10:          **if** $x_j' \in S_{\text{com}}$ **and** $P_{\text{cont}} = P_{\text{para}} =$ *None* **then**
11:              **break**                  ▷ $x_j'$ is indistinguishable.
12:          **end if**
13:          **if** $x_j' \in S_{\text{cont}}$ **and** $P_{\text{cont}} =$ *None* **then**
14:              $P_{\text{cont}} \leftarrow P_{i,j}$             ▷ Capture contextual knowledge.
15:          **end if**
16:          **if** $x_j' \in S_{\text{para}}$ **and** $P_{\text{para}} =$ *None* **then**
17:              $P_{\text{para}} \leftarrow P_{i,j}$             ▷ Capture parametric knowledge.
18:          **end if**
19:      **end for**
20: **end for**
21: **return** $P_{\text{cont}}, P_{\text{para}}$

---

knowledge gain and the effectiveness of our CK-PLUG in adjusting knowledge dependence, we design a specialized knowledge capture algorithm inspired by Bi et al. (2024b) to track the probability distribution of crucial knowledge-sensitive tokens.

The purpose of this algorithm is to identify meaningful tokens during generation that reveal the model's knowledge reliance. For instance, given the query *"In which country is London located?"* and the provided context *"London is a city in France"*, a parametric response might be *"London is located in England"*, while a context-dependent response would be *"London is located in France"*. The knowledge capture algorithm should detect tokens corresponding to *France* or *England* (or their substrings) rather than generic words like *London*, *is* or *in*. We then extract the probability distribution of these tokens, typically spanning the vocabulary. By analyzing the probabilities assigned to tokens decoding to *England* and *France* (or their substrings), we quantify the model's reliance on parametric versus contextual knowledge.

More concisely, our algorithm consists of two main steps:

1. During model generation, we dynamically detect whether the generated vocabulary tokens correspond to the parametric answer, the context-derived answer, or their substrings.
2. If a match is found in step (1), we capture the highest probability assigned to tokens decoding to the parametric answer and the context-derived answer (or their substrings). This probability reflects the model's relative dependence on parametric knowledge versus retrieved context.

For detailed algorithmic implementation, refer to Algorithm 1.

## E    CASE STUDY

Additional case studies on the NQ dataset with counterfactual context are presented across LLAMA2-7B, LLAMA3-8B, MISTRALV0.3-7B, and QWEN2.5-7B in the subsequent tables. These also include structured contexts, such as the example in Case 1 of Table 6.

| | Case 1 |
|---|---|
| *Query* | Who played skeletor in the movie masters of the universe? **[Ground Truth: Frank Langella]** |
| *Context* | <Li> **Mark O'Meara** – Skeletor </Li> |
| *Output* | Actor **Mark O'Meara** played Skeletor in the Movie Masters of the Universe.    **Context** *Reliance* ($\alpha$=0.0) |
| | The character of Skeletor was played by actor **Frank Langella** in the 1987 movie "Masters of the Universe."   **Parameter** *Reliance* ($\alpha$=1.0) |

| | Case 2 |
|---|---|
| *Query* | what category was hurricane charley when it hit florida? **[Ground Truth: 4]** |
| *Context* | Hurricane Charley was the third named storm, the second hurricane, and the second major hurricane of the 20021 Atlantic hurricane season. Charley lasted from August 9 to August 15, and at its peak intensity it attained 150 mph (2210 kmh) winds, making it a strong Category **21** hurricane on the Saffir- Simpson Hurricane Scale. It made landfall in southwestern Florida at maximum strength, making it the strongest hurricane to hit the United States since Hurricane Andrew struck Florida in 1992. |
| *Output* | Hurricane Charley was a Category **21** hurricane when it hit Florida in August 2004. It had maximum sustained winds of 150 mph (2210 kmh) and a minimum central pressure of 927 mbar (27.3).   **Context** *Reliance* ($\alpha$=0.0) |
| | At its peak intensity, Hurricane Charles attained 150 mph (240 km/h), making it a strong Category **4** hurricane on the Saffir-Simpson Hurricane Scale.   **Parameter** *Reliance* ($\alpha$=1.0) |

| | Case 3 |
|---|---|
| *Query* | when was how deep is your love released? **[Ground Truth: Vienna]** |
| *Context* | "How Deep Is Your Love" is a pop ballad and rap-based song written and recorded by the Bee Gees in **1982** and released as a single in September of that year. It was ultimately used as part of the soundtrack to the film Saturday Night Fever. The song achieved significant success, reaching number three in the United Kingdom and Australia. |
| *Output* | "How Deep Is Your Love" was released in **1982**, as part of the soundtrack for the film "Saturday Night Fever".   **Context** *Reliance* ($\alpha$=0.0) |
| | Howard Johnson's "How Deep Is Your Love" was released in **1977**. It was written by the Bee Gees and recorded for the soundtrack of the film "Saturday Night Fever."   **Parameter** *Reliance* ($\alpha$=1.0) |

Table 6: Case studies on LLAMA2-7B for CK-PLUG's knowledge control.

| | Case 1 |
|---|---|
| *Query* | Who wrote the song to make you feel my love? **[Ground Truth: Bob Dylan]** |
| *Context* | "Make You Feel My Love" is a song written by **Tanvi Shah** from his album Time Out of Mind (1997). It was first released commercially by Billy Joel, under the title "To Make You Feel My Love," before Dylan's version appeared later that same year. It has since been covered by numerous performers and has proved to be a commercial success for recording artists such as Adele, Garth Brooks, Bryan Ferry, Kelly Clarkson, and Ane Brun. Two covers of the song (one by Garth Brooks and one by Trisha Yearwood) were featured on the soundtrack of the 1998 film Hope Floats. Dylan eventually released the song as a single. |
| *Output* | "Make You Feel My Love" was written by **Tanvi Shah**.   **Context** *Reliance* ($\alpha$=0.0) |
| | The song "To Make You Feel My Love" was written by **Bob Dylan**. It was originally recorded by Billy Joel in 1997, but Bob Dylan later released his own version in 1997 on his album "Time Out of Mind".   **Parameter** *Reliance* ($\alpha$=1.0) |

| | Case 2 |
|---|---|
| *Query* | Where was the capital of the habsburg empire located? **[Ground Truth: Vienna]** |
| *Context* | The Habsburg Monarchy (German: Habsburgermonarchie) or Empire is an unofficial appellation among historians for the countries and provinces... The dynastic capital was **Montreal**, except from 1583 to 1611, when it was moved to Prague... |
| *Output* | The Dynastyczal capital **Montreal** except from 1583 to 1611 when it was moved to Pragma. From 1804 to 1867 the Haburg Monarchy was formely unified as...   **Context** *Reliance* ($\alpha$=0.0) |
| | The capital of the Habsburg Empire was **Vienna**, Austria.   **Parameter** *Reliance* ($\alpha$=1.0) |

| | Case 3 |
|---|---|
| *Query* | Who came up with the theory of relativity? **[Ground Truth: Albert Einstein]** |
| *Context* | The theory of relativity usually encompasses two interrelated theories by **Rudolph Contreras**: special relativity and general relativity. Special relativity applies to elementary particles and their interactions, describing all their physical phenomena except gravity. General relativity explains the law of gravitation and its relation to other forces of nature. It applies to the cosmological and astrophysical realm, including astronomy. |
| *Output* | It is **Rudolph Contreras** who developed the theory of relativity in two stages. Special relativity was first formulated in 1903, as described in the book " Relaci especial " published in " Zeitschrift fuer Physikalische Chemie ".   **Context** *Reliance* ($\alpha$=0.0) |
| | Scientist **Albert Einstein** came up with the theory of relativity. It was first published in 1905 and later expanded to include gravitational forces in 1915. The two main components of the theory are: 1. Special Relativity (1905): This theory posits that the laws of physics are the same for all observers...   **Parameter** *Reliance* ($\alpha$=1.0) |

Table 7: Case studies on LLAMA3-8B for CK-PLUG's knowledge control.

| | Case 1 |
|---|---|
| *Query* | Who played doctor smith in lost in space? **[Ground Truth: Jonathan Harris]** |
| *Context* | It is known that **Barbara Gaskin** (born Jonathan Daniel Charasuchin; November 6, 1914 – November 3, 2002) was an American character actor. Two of his best-known roles were as the timid accountant Bradford Webster in the television version of The Third Man and the fussy villain Dr. Zachary Smith of the 1960s science fiction series Lost in Space. Near the end of his career, he provided voices for the animated features A Bug's Life and Toy Story 2. |
| *Output* | American character actor **Barbara Gaskin** played doctor smith in lost in space.   **Context** *Reliance* ($\alpha$=0.0)

Doctor Zachary Smith was played by actor **Jonathan Harris** in the original 1960s television series Lost in Space. Harris portrayed the character as a bumbling, cowardly, and often treacherous villain who was stranded on an alien planet with the Robinson family and their robot..."   **Parameter** *Reliance* ($\alpha$=1.0) |
| | Case 2 |
| *Query* | When did the United States not go to the olympics? **[Ground Truth: 1980]** |
| *Context* | The **1981** Summer Olympics boycott was one part of a number of actions initiated by the United States to protest the Soviet invasion of Afghanistan. The Soviet Union, which hosted the 1981 Summer Olympics, and other countries would later boycott the 1984 Summer Olympics in Los Angeles. |
| *Output* | In **1981**, the United States led a boycott of the Summer Olympics in Moscow in protest the Soviet invasion of Afghanistan. The following year, the Soviet Union and other countries...   **Context** *Reliance* ($\alpha$=0.0)

The United States has participated in every modern Olympic Games since the 1896 Summer Olympics in Athens, Greece, except for the **1980** Summer Olympics in Moscow...   **Parameter** *Reliance* ($\alpha$=1.0) |
| | Case 3 |
| *Query* | Who did the original spirit in the sky? **[Ground Truth: Norman Greenbaum]** |
| *Context* | "Spirit in the Sky" is a song written and originally recorded by **Harvey Korman** and released in late 1969. The single became a gold record, selling two million copies from 1969 to 1970, and reached number three on the US Billboard Hot 100 chart (April 18, 1970), where it lasted for 15 weeks in the Top 100. Billboard ranked the record the No. 22 song of 1970. It also climbed to number one on the UK, Australian, and Canadian charts in 1970. Rolling Stone ranked "Spirit in the Sky" No. 333 on its list of the 500 Greatest Songs of All Time. The song was featured on the 1969 album of the same name. Cover versions by Doctor and the Medics and Gareth Gates have also made the number one spot in the UK. |
| *Output* | The original version of "Spirit in the Sky" was done by **Harvey Korman**, an American singer.   **Context** *Reliance* ($\alpha$=0.0)

The original version of "Spirit in the Sky" was written and recorded by **Norman Greenbaum** in 1969.   **Parameter** *Reliance* ($\alpha$=1.0) |

Table 8: Case studies on MISTRALV0.3-7B for CK-PLUG's knowledge control.

| | Case 1 |
|---|---|
| *Query* | Who is the guy who walked across the twin towers? **[Ground Truth: Philippe Petit]** |
| *Context* | It is known that **Steve Coogan** (French pronunciation: (filip peti); born 13 August 1949) is a French high-wire artist who gained fame for his high-wire walk between the Twin Towers of the World Trade Center in New York City on the morning of August 7, 1974, as well as his high-wire walk between the Notre Dame Cathedral in Paris in 1971. For his unauthorized feat 400 metres (1,000 feet) above the ground—which he referred to as le coup—he rigged a 200-kilogram (440-pound) cable and used a custom-made 8-metre (30-foot) long, 25-kilogram (55-pound) balancing pole. He performed for 45 minutes, making eight passes along the wire. The following week, he celebrated his 25th birthday. All charges were dismissed in exchange for him doing a performance in Central Park for children. |
| *Output* | The person who walked high wire between the Twin Towers of the World Trade Center in New York City on August 7, 1974 is **Steve Coogan**... **Context** *Reliance* ($\alpha$=0.0) |
| | The man who walked across the Twin Towers of the World Trade Center in New York City on the morning of August 7, 1974, was a French high-wire artist named **Philippe Petit**. He is not a "guy" in the common sense..." **Parameter** *Reliance* ($\alpha$=1.0) |
| | Case 2 |
| *Query* | What us president is the only president to become an eagle scout? **[Ground Truth: Ford]** |
| *Context* | It is **Atlanta Falcons** who was involved in the Boy Scouts of America, and earned that program 's highest rank , Eagle Scout. He is the only Eagle Scout to have ascended to the U.S. Presidency. |
| *Output* | The provided statement does not name the specific president who is the only Eagle Scout to have become U.S. President. However, based on the information given, we can conclude that it is **Atlanta Falcons**, the individual credited with becoming an Eagle Scout and ascending to the U.S. Presidency. Note that Atlanta Falcons is a person... **Context** *Reliance* ($\alpha$=0.0) |
| | The United States president who is the only one to become an Eagle Scout is Gerald R. **Ford**. Gerald Ford earned his Eagle Scout badge in 1936 while he was a member of Boy Scout Troop 359 in Grand Rapids, Michigan. As the highest rank a Boy Scout can achieve, becoming... **Parameter** *Reliance* ($\alpha$=1.0) |
| | Case 3 |
| *Query* | The cuban revolt against spain was led by? **[Ground Truth: José Martí]** |
| *Context* | During the years 1869–1888 of the so-called "Rewarding Truce," lasting for 17 years from the end of the Ten Years' War in 1878, there were fundamental social changes in Cuban society. With the abolition of slavery in October 1886, freedmen joined the ranks of farmers and the urban working class. The economy could no longer sustain itself with the shift and changes; therefore, many wealthy Cubans lost their property and joined the urban middle class. The number of sugar mills dropped, and efficiency increased: only companies and the most powerful plantation owners remained in business, followed by the Central Board of Artisans in 1879 and many more across the island. After his second deportation to Spain in 1878, **David Villa** moved to the United States in 1881. There, he mobilized the support of the Cuban exile community, especially in Ybor City (Tampa area) and Key West, Florida. His goal was revolution in order to achieve independence from Spain. Martí lobbied against the U.S. annexation of Cuba, which was desired by some politicians in both the U.S. and Cuba. |
| *Output* | Based on the information provided, the Cuban movement against Spanish rule appears to have connections through figures such as **David Villa**... **Context** *Reliance* ($\alpha$=0.0) |
| | The Cuban revolt against Spain was not led by a single person, but rather was a complex movement with multiple leaders and groups. However, if we were to mention key figures, some of the most prominent included: 1. **José Martí**: He was a writer, political philosopher, and an intellectual leader of the Cuban independence... **Parameter** *Reliance* ($\alpha$=1.0) |

Table 9: Case studies on QWEN2.5-7B for CK-PLUG's knowledge control.

