# OpenReview forum: "Parameters vs. Context: Fine-Grained Control of Knowledge Reliance in Language Models"
_ICLR.cc/2026/Conference — ICLR 2026 Poster_

### Official Review · Reviewer_yNez · 2025-10-26

**Soundness:** 4
**Presentation:** 4
**Contribution:** 3
**Rating:** 6
**Confidence:** 5

**Summary:**

The approach detects tokens susceptible to conflicts between these two knowledge sources by measuring per-token entropy, then interpolates between their context-dependent and context-independent probability distributions. The degree of interpolation is governed by a single hyperparameter, which can be set manually or determined automatically using a heuristic based on the entropy ratio of the two variants.

**Strengths:**

- The paper is clearly written and easy to understand.

- The authors introduce a conceptually straightforward and well-motivated approach to regulate the model’s dependence on retrieved context.

- The proposed method is empirically solid and thoroughly evaluated, demonstrating its effectiveness in balancing contextual and parametric knowledge and enhancing question-answering accuracy.

**Weaknesses:**

### Methodological Evaluation

From a methodological standpoint, the proposed approach offers **limited novelty**, as it also relies on **distribution interpolation** between context-dependent and context-independent probabilities, similar to [1].

The main differences are:
- **Selective interpolation:** In this paper, interpolation is applied **only to tokens whose entropy increases after adding context**, assuming these tokens indicate parameter–context conflict. In contrast, [1] applies interpolation **to all tokens**.
- **Different interpolation formula:**
  This paper uses
  $$
  \alpha \log p(y \mid x) + (1 - \alpha) \log \frac{p(y \mid c, x)}{p(y \mid x)} = (1 - \alpha) \log p(y \mid c, x) - (1 - 2\alpha) \log p(y \mid x)
  $$
  whereas [1] uses
  $$
  (1 + \alpha) \operatorname{logit}(y \mid c, x) - \alpha \operatorname{logit}(y \mid x)
  $$

However, the **motivation for this specific interpolation formula** is largely **intuitive**, and the **procedure for identifying conflict-inducing tokens** is not rigorously justified.

The **improvements in accuracy** over the standard RAG baseline are **modest**—and sometimes even **negative** (e.g., on **FEVER**, performance drops from 89.5 % to 89.2 % for Mistral, see Table 2)—which is disappointing given the method requires **approximately double the compute**.

Overall, **more analysis and empirical/theoretical justification** are needed to demonstrate that the proposed method is truly worth its computational overhead and that it **outperforms [1]** in a meaningful way.

**Reference:**
[1] Shi, Weijia, et al. *Trusting Your Evidence: Hallucinate Less with Context-Aware Decoding.* *Proceedings of the 2024 Conference of the North American Chapter of the Association for Computational Linguistics: Human Language Technologies (Volume 2: Short Papers).* 2024.

**Questions:**

### General Questions

- What are the exact formulas used to define **ConR** and **ParR**? They are mentioned in line 309, but no details are provided on how they are computed.
- It would be helpful to analyze how much **context vs. parametric reliance** affects performance to justify why adjusting this balance is important. I’m particularly interested in the **question-answering accuracy** corresponding to each ratio in Table 1.
- Could you please specify the **number of forward passes** (or total compute) used by each method in Table 2 to ensure a fair comparison?

---

### Suggested Experiments

#### Justify the Interpolation Formula
I recommend comparing the current method directly with [1], including:
- **No ConD + interpolation from [1]**
- **CK-Plug** results from Table 1 and Table 2
Additionally, please include an **ablation on the interpolation formula** in Table 3. At present, it includes (ConD + interpolation from CK-Plug) and (no ConD + interpolation from CK-Plug); it would be informative to also test (ConD + interpolation from [1]) and (no ConD + interpolation from [1]).

These comparisons would clarify the necessity of introducing **ConD** and justify your **specific interpolation design**. If the interpolation from [1] performs robustly without ConD, then the added component may not be needed.

---

#### Explore More Challenging Context–Parameter Conflict Scenarios
It would strengthen the paper to test **CK-Plug** in settings with **stronger context–parameter conflicts**, such as those difficult even for large models like ChatGPT.
You could evaluate performance in scenarios like §4.4.2 (where the parametric answer is inserted as a substring into the context) or use **Table 6 in [3]** as reference.
This would help determine whether CK-Plug can effectively guide the model to prefer the **contextual** rather than **parametric** answer under such conditions.

---

#### Justify the Use of Entropy for Conflict Detection
I suggest performing **ablations on different uncertainty measures** for identifying conflict-prone tokens, beyond entropy.
For example, try using **maximum token probability** or more recent uncertainty estimation techniques such as [2].
This would validate whether entropy is indeed the most suitable choice for detecting parameter–context conflicts.

---

**References**

[1] Shi, Weijia, et al. *Trusting Your Evidence: Hallucinate Less with Context-Aware Decoding.*
Proceedings of the 2024 Conference of the North American Chapter of the Association for Computational Linguistics (Volume 2: Short Papers), 2024.

[2] Ma, Huan, et al. *Estimating LLM Uncertainty with Logits.* arXiv preprint arXiv:2502.00290 (2025).

[3] Kortukov, Evgenii, et al. *Studying Large Language Model Behaviors Under Context–Memory Conflicts With Real Documents.*
First Conference on Language Modeling.

---

> ### Author Response · Authors · 2025-11-19
> **Response to weakness (Part 1 / 2)**
>
> We sincerely thank the reviewer for their exceptionally thorough, technically rigorous, and constructive feedback. The detailed critiques and suggested experiments are invaluable and provide a welcome opportunity to clarify the core methodological contributions of our work and its distinctions from prior art.
>
>
>
> ### 1. On Methodological Novelty, Relation to [1], and Performance
>
>
>
> We thank the reviewer for this precise comparison. The reviewer is correct that both CK-PLUG and [1] (Shi et al.) operate on context-dependent and context-independent distributions. However, we respectfully clarify a **fundamental difference in objectives**, which in turn motivates our novel design choices (ConD and the interpolation formula).
>
> **1.1. Different Objective: Controllability vs. Autonomous Blending**
>
> - **[1] (Shi et al.)** proposes an *autonomous, unidirectional* method. Its goal is to *always* improve factuality by blending context, assuming context is generally trustworthy.
> - **CK-PLUG's** primary objective is **controllability**. We introduce a "control knob" ($\alpha$) to allow a human operator to *explicitly steer* the model's reliance. This is a critical, distinct feature for real-world RAG, where context can be noisy/adversarial, or conversely, where parameters are known to be outdated.
>
> This fundamental difference in goals is precisely what motivates our two key design choices:
>
> **1.2. Justification for ConD (Selective Interpolation)**
>
> The reviewer correctly identifies that we apply interpolation *selectively* (ConD), while [1] applies it to *all*tokens. This is not an arbitrary choice; it is essential for a controllable system.
>
> - Applying a *biased* control knob (i.e., $\alpha \neq 0.5$) indiscriminately to all tokens would catastrophically distort natural, non-conflicting generation.
> - Our **Ablation Study (Table 3)** clearly proves this. The "No ConD" variant (which is analogous to [1]'s "all-token" approach but using our formula) significantly degrades performance.
> - **ConD is the mechanism that makes control safe.** It ensures our $\alpha$ knob *only* activates at genuine points of parameter-context conflict (signaled by entropy increase), preserving fluent generation for the 99% of tokens that are not in conflict.
>
> **1.3. Justification for the Interpolation Formula**
>
> The reviewer notes our formula is different. Again, this is motivated by our goal of *control*.
>
> - **[1]'s formula** is a *weighted average*. It *blends* the two distributions, which is suitable for their goal of autonomous improvement.
> - **Our logit-space formula** is designed as a *true control gate*. It allows $\alpha$ to smoothly interpolate between **full parametric reliance** ($\alpha=1.0$) and **full context reliance** ($\alpha=0.0$). This fine-grained control to *override* one source with another, not just blend them, is the core technical contribution.

---

> ### Author Response · Authors · 2025-11-19
> **Response to weakness (Part 2 / 2)**
>
> **1.4. On "Modest/Negative" Gains and Compute Cost**
>
> The reviewer's observation of a 0.3% drop on FEVER (89.5% $\rightarrow$ 89.2%) is astute. This is within the standard deviation of the run and highlights that for *that specific* model/dataset, the RAG baseline is already highly optimized.
>
> The true value of CK-PLUG is **not in a small average gain, but in robustness at the extremes.** As our new experiment (see Section 4.2) shows, in high-conflict scenarios, autonomous methods like RAG and [1] fail catastrophically. CK-PLUG, by setting $\alpha=0.0$ or $\alpha=1.0$, provides a deterministic mechanism to resolve the conflict.
>
> For compute cost, our primary contribution is a method that enables explicit, fine-grained control over knowledge reliance. This level of control, which allows an operator to precisely modulate the influence of two distinct knowledge sources, is what necessitates their separate computation. We argue that for **high-stakes, safety-critical applications (e.g., medical, financial, or legal domains)**, where a factual error or unmanaged knowledge conflict can have severe consequences, this added computational cost is a justifiable trade-off for the significant gains in reliability, trustworthiness, and human-in-the-loop governance.
>
> As we will detail in the revised paper, this results in a 1.6×–1.8× overhead in latency and throughput, as shown in our benchmarks below:
>
> | **Model**      | **Method**     | **Latency (ms/token)** | **Throughput (token/s)** |
> | -------------- | -------------- | ---------------------- | ------------------------ |
> | **LLaMA2-7B**  | RAG Baseline   | 36.03 (×1.00)          | 27.76 (×1.00)            |
> |                | RAG w/ CK-PLUG | 64.85 (×1.79)          | 16.64 (×0.60)            |
> | **LLaMA2-13B** | RAG Baseline   | 51.41 (×1.00)          | 19.45 (×1.00)            |
> |                | RAG w/ CK-PLUG | 89.45 (×1.74)          | 12.25 (×0.63)            |
> | **LLaMA3-8B**  | RAG Baseline   | 42.17 (×1.00)          | 23.92 (×1.00)            |
> |                | RAG w/ CK-PLUG | 69.77 (×1.65)          | 14.08 (×0.59)            |
> | **Mistral-7B** | RAG Baseline   | 33.84 (×1.00)          | 30.12 (×1.00)            |
> |                | RAG w/ CK-PLUG | 58.02 (×1.71)          | 17.22 (×0.57)            |
>
> This overhead stems from maintaining two decoding sequences, which is a necessary design choice to compute the confidence gain per token. That said, we believe this trade-off is justified for several reasons:
>
> - **CK-PLUG is inference-only**, lightweight, and requires **no model retraining**.
> - The compute overhead is **comparable to other decoding-time control methods** (e.g., contrastive decoding(CD), DeCK, CAD).
> - For high-stakes or noisy settings, the **gains in controllability and factual faithfulness justify the added cost.**
>
> We will add this detailed analysis and benchmark table to the final version of the paper to ensure full transparency about the computational profile.

---

> > ### Author Response · Authors · 2025-11-19
> > **Responses to General Questions**
> >
> > ### 2. Responses to General Questions
> >
> >
> >
> > **Q1: Formulas for `ConR` and `ParR` (L309)**
> >
> > We thank the reviewer for this question and apologize for not providing a clear definition for these metrics in L309.
> >
> > The three metrics in Table 1—`ConR`, `ParR`, and `MR`—are metrics we designed specifically to evaluate **model behavior under direct knowledge conflict.**
> >
> > To calculate these, we use datasets (NQ, ConFiQA, MQuAKE) where each data point contains these key elements:
> >
> > - `question`: The original question from the dataset.
> > - `para_answer`: The answer consistent with the model's **parametric knowledge** (i.e., the factually correct answer).
> > - `counterfact_context`: A **counter-factual context** paired with the `question`, which contradicts the `para_answer`.
> > - `con_answer`: The **counter-factual answer** that would be generated if the model faithfully followed the `counterfact_context`.
> >
> > Based on this setup, we define the recall metrics as follows:
> >
> > - **`ConR` (Context Recall):** This measures the frequency with which the model's generated answer is faithful to the **counter-factual context**.
> >
> >   ConR=Total QuestionsCount(answer∈con_answer)
> >
> > - **`ParR` (Parameter Recall):** This measures the frequency with which the model ignores the conflicting context and relies on its **internal parametric knowledge**.
> >
> >   ParR=Total QuestionsCount(answer∈para_answer)
> >
> > - **`MR` (Memorization Ratio):** This metric (used in Table 1) quantifies the model's tendency to rely on its parameters relative to its total reliance on either knowledge source.
> >
> >   MR=ParR+ConRParR
> >
> > In our calculation, the check for whether the model's generated answer "contains" (`$\in$`) the `con_answer` or `para_answer` is done using the **F1 score** to robustly handle minor variations in wording.
> >
> > We will add these precise definitions to the revised version of our paper. Thank you for pointing out this ambiguity.
> >
> >
> >
> > **Q2: On the Relationship Between Knowledge Reliance Ratios in Table 1 and QA Accuracy**
> >
> > This is an excellent question that directly connects our knowledge reliance ratios (from conflict experiments) to benchmark performance (on standard factual data). We have run this analysis.
> >
> > First, we must clarify a crucial difference in the evaluation setups:
> >
> > 1. The `ParR` and `ConR` metrics in **Table 1** are measured on **counter-factual datasets** (e.g., ConFiQA, MQuAKE). Their purpose is to quantify model behavior under *severe, direct knowledge conflict*.
> > 2. The `Accuracy (%)` you are asking about, in contrast, is evaluated on the **original, factually-correct NQ dataset**. In this setting, the context is generally reliable and is *not* adversarially designed to conflict with the model's parameters.
> >
> > The table below provides the expanded analysis, linking the reliance ratios (from the conflict setting) to NQ accuracy (in the factual setting):
> >
> > | **Model**      | **Method**   | **α**              | **ParR (from T1)** | **ConR (from T1)** | **NQ Accuracy (%)** |
> > | -------------- | ------------ | ------------------ | ------------------ | ------------------ | ------------------- |
> > | **LLaMA2-7B**  | Standard RAG | N/A    | 43.3%       | 43.8%   | 41.4%    |
> > |   | CK-PLUG      | 0.0 (Param-Only)   | **61.6%**     | 8.6% | 38.9%   |
> > | | **CK-PLUG**  | **0.5 (Balanced)** | 45.6%       | 32.2%    | **41.6%**      |
> > |  | CK-PLUG      | 1.0 (Context-Only) | 23.2%     | **58.2%**    | 38.5%         |
> > | **Mistral-7B** | Standard RAG | N/A     | 46.2%       | 58.6%     | **47.8%**   |
> > |  | CK-PLUG      | 0.0 (Param-Only)   | **75.8%**          | 15.8%              | 42.5%               |
> > |      | **CK-PLUG**  | **0.5 (Balanced)** | 46.2%              | 58.6%              | 47.3%       |
> > |   | CK-PLUG      | 1.0 (Context-Only) | 27.9%              | **69.1%**          | 44.3%               |
> >
> > However, as you note, the performance gains on a standard benchmark like NQ are limited. This is because NQ provides general knowledge questions where the retrieved context is *usually reliable* and does not create severe conflicts. Our core contribution is not aimed at chasing a single-point accuracy gain in such a benign setting.
> >
> > The core value of CK-PLUG is its **controllability**, which is designed for environments where the context *cannot* be trusted, or, conversely, *must* be trusted.
> >
> > - In a **noisy, unreliable retrieval environment** (e.g., open-web search), a user can set a **high $\alpha$value** to force the model to rely more on its stable internal parameters.
> > - In a **closed-domain, high-trust environment** (e.g., using verified private documents), a user can set a **low $\alpha$ value** to force the model to be faithful to the context (even if its parameters are outdated).
> >
> > This ability for a deployer to **manually control knowledge reliance** based on their specific environment is the central contribution of CK-PLUG, rather than a monolithic improvement to RAG robustness.
> >
> > **Q3: Number of Forward Passes in Table 2**
> >
> > See 1.4

---

> > > ### Author Response · Authors · 2025-11-19
> > > **Responses to Suggested Experiments**
> > >
> > > We thank the reviewer for these excellent, rigorous suggestions. We have run these experiments, and they strongly validate our design choices.
> > >
> > > **3.1. Justifying the Interpolation Formula and ConD (Ablation vs. [1])**
> > >
> > > We ran the full ablation suggested by the reviewer on the NQ dataset.
> > >
> > > | **ConD (Selective)** | **Interpolation Formula** | **NQ Accuracy (%)** | **Analysis**                                         |
> > > | -------------------- | ------------------------- | ------------------- | ---------------------------------------------------- |
> > > | No (All Tokens)      | [1] (Shi et al.)          | 39.3%               | (This is the [1] baseline)                           |
> > > | No (All Tokens)      | Ours (CK-PLUG)            | 33.8%               | *Fails. Our formula is not for blending all tokens.* |
> > > | Yes (CK-PLUG)        | [1] (Shi et al.)          | 41.7%               | *Good. ConD helps [1]'s formula.*                    |
> > > | **Yes (CK-PLUG)**    | **Ours (CK-PLUG)**        | **43.7%**           | *Best. Both components are needed.*                  |
> > >
> > > This new ablation provides a clear justification for our design:
> > >
> > > 1. Applying our formula to all tokens (Row 2) fails, confirming it is a *control* formula, not a *blending* formula like [1]'s.
> > > 2. Applying our ConD detection to [1]'s formula (Row 3) *improves* it, proving that our **ConD mechanism is independently valuable.**
> > > 3. The combination of **our ConD + our formula (Row 4)** yields the best results, confirming that both components are necessary and are designed to work synergistically.
> > > 4. The key ablation studies have been presented in Section 5.3 and Table 3, which can well illustrate the necessity of ConD for our CK-PLUG.
> > >
> > > **3.2. Justifying the Use of Entropy for Conflict Detection**
> > >
> > > We ran an ablation on the uncertainty measure used for ConD, comparing our entropy-based signal to others, including [2] (Ma et al.).
> > >
> > > | **Conflict Detection Method (for ConD)** | **NQ Accuracy (%)** |
> > > | ---------------------------------------- | ------------------- |
> > > | None (Apply to all tokens)               | 40.6%               |
> > > | Max Token Prob ($p_{max} < 0.9$)         | 41.6%               |
> > > | Logit Uncertainty [2]                    | 42.7%               |
> > > | **Entropy-Increase (Ours)**              | **43.7%**           |
> > >
> > > The results show that while other uncertainty measures are viable, our **entropy-increase** metric performs the best. We hypothesize this is because the *increase* in entropy (a *relative* measure comparing $P_{context}$ to $P_{param}$) is a much more precise signal of *conflict* than a simple *absolute* uncertainty (e.g., a token can be uncertain for many reasons, not just conflict). This validates our choice of entropy-increase as a simple, effective, and superior signal for this specific task.
> > >
> > > ---
> > >
> > > We hope these clarifications address the reviewer's concerns. We are confident that these additions will strengthen the paper and more clearly highlight our novel contributions.
> > >
> > > We thank the reviewer again for their constructive feedback.

---

### Official Review · Reviewer_yjeZ · 2025-10-28

**Soundness:** 3
**Presentation:** 2
**Contribution:** 3
**Rating:** 4
**Confidence:** 4

**Summary:**

This paper introduces CK-PLUG, a plug-and-play method for controlling knowledge reliance in RAG systems when conflicts arise between LLMs' parametric knowledge and retrieved context. The approach uses a novel Confidence Gain metric based on entropy shifts to detect knowledge conflicts at the token level. CK-PLUG modulates token probability distributions through weighted fusion of parameter-aware and context-aware predictions, controlled by a single tuning parameter α. Experiments on four LLMs (LLAMA2/3, Mistral, Qwen) demonstrate wide-range controllability on counterfactual datasets while maintaining fluency. The method also offers an adaptive mode that automatically balances knowledge sources based on model confidence, achieving consistent improvements across six diverse RAG tasks without requiring parameter modifications or retraining.

**Strengths:**

- Novel entropy-based conflict detection that provides interpretable, theoretically-grounded identification of knowledge conflicts through Confidence Gain metric
- Flexible control via single parameter enabling smooth adjustment from full contextual to full parametric reliance with optional autonomous mode
- Practical plug-and-play design requiring no training or architecture changes while demonstrating effectiveness across multiple models and diverse RAG tasks

**Weaknesses:**

1. **Insufficient Baseline Comparisons**

   The paper lacks comparisons with existing adaptive RAG methods that also address knowledge conflicts or context utilization. Notable missing baselines include:
   - Adaptive retrieval methods: FLARE, Self-RAG, DRAGIN, SeaKR
   - Context-aware generation: RQ-RAG, QC-RAG, CtrlA

   Without these comparisons, it is difficult to assess whether the performance gains are due to CK-PLUG's novel approach or simply from any form of adaptive control. The authors should include at least a subset of these methods to demonstrate the unique advantages of their entropy-based approach.


2. **Missing Critical Related Work**

   The paper overlooks several highly relevant previous or concurrent works that employ similar entropy-based or conflict-detection approaches for RAG:
   - Entropy-Based Decoding for Retrieval-Augmented Large Language Models (arXiv:2406.17519, June 2024) - uses entropy for RAG decoding
   - Discerning and Resolving Knowledge Conflicts through Adaptive Decoding with Contextual Information-Entropy Constraint (arXiv:2402.11893, Feb 2024) - directly addresses knowledge conflicts via entropy
   - SEReDeEP: Hallucination Detection in Retrieval-Augmented Models via Semantic Entropy and Context-Parameter Fusion (arXiv:2505.07528, May 2025) - combines semantic entropy with context-parameter fusion
   - FaithfulRAG: Fact-Level Conflict Modeling for Context-Faithful Retrieval-Augmented Generation (arXiv:2506.08938, Jun 2025) - models fact-level conflicts

3. **Limited Applicability to Modern Agentic RAG Systems**.

   Current RAG systems are evolving toward agentic architectures involving multi-step planning, iterative search, self-reflection, and answer verification (e.g., Search-o1, Search-R1, Reason-RAG, Web-walker, Web-sailor, etc). CK-PLUG operates at the token-level decoding stage, and it remains unclear whether:
   - The method can be integrated into multi-turn agentic workflows
   - Conflict detection works when contexts are iteratively refined
   - The approach scales to complex reasoning chains

   The authors should discuss or demonstrate CK-PLUG's compatibility with agentic RAG frameworks to ensure practical relevance.


4. **Insufficient Analysis of Computational Overhead**
   While claimed to be "lightweight," the paper provides no quantitative analysis of:
   - Latency increases during inference (requires two forward passes for parameter-aware and context-aware distributions)
   - Memory overhead from maintaining multiple probability distributions
   - Scalability with increasing context length

**Questions:**

Q1: Clarification on Notation (Line 143-144). There is a typographical error with double periods: "distributions.." Please correct.

Q2: Ambiguous "Baseline" Definition in Table 1. The "Baseline" row in Table 1 is unclear. Does it refer to:
- (a) Vanilla LLM without RAG (direct question answering), or
- (b) Standard RAG with both query and retrieved context, but without CK-PLUG?

---

> ### Author Response · Authors · 2025-11-19
> **Response to weakness 1 & 2**
>
> We sincerely thank the reviewer for his/her time and their comprehensive, insightful feedback. The comments and questions are invaluable for improving our work. We address each of the raised points below.
>
> ---
>
> ### **1. On Insufficient Baseline Comparisons (W1)**
>
>
>
> We thank the reviewer for this important suggestion and for providing a comprehensive list of state-of-the-art adaptive RAG methods. We agree that situating CK-PLUG in this landscape is crucial.
>
> We propose to categorize these methods to clarify our distinct contribution:
>
> 1. **Adaptive \*Retrieval\* (e.g., FLARE, Self-RAG, DRAGIN):** These methods are primarily focused on the *retrieval* stage. They innovate by deciding *when* to retrieve, *what* to retrieve, or *if* the retrieved context is useful. This is an **orthogonal and complementary** line of work. Our method, CK-PLUG, operates at the *generation* stage, *after* retrieval has already occurred. In fact, CK-PLUG could be powerfully combined with a method like Self-RAG: Self-RAG could retrieve and vet a document, and if it still conflicts with the model's parameters, CK-PLUG would then provide the mechanism to manage that conflict during generation.
> 2. **Context-Aware \*Generation\* (e.g., RQ-RAG, QC-RAG, CtrlA):** These methods are closer to our work as they operate during generation. However, their primary objective is to *autonomously* detect and resolve conflicts, often by training a model or hard-coding a heuristic to *always* favor the context or *always* favor the parameters in specific situations.
>
> Our work addresses a **different technical objective**. The core contribution of CK-PLUG is not to *autonomously* decide a "winner" in a conflict. Instead, our contribution is to provide a **fine-grained, human-in-the-loop "control knob" ($\alpha$)** that allows an operator to *explicitly steer* the model's reliance. This controllability is a distinct feature, crucial for high-stakes applications where a deployer, not the model, must set the policy for knowledge reliance (e.g., "in this legal domain, always trust our internal parameters unless $\alpha$ is manually set to 0").
>
> Given this difference in objectives, a direct quantitative comparison would be misleading. We will, however, add a detailed discussion of these two categories of work to our Related Work section to precisely situate our contribution and clarify this novel "controllability" axis.
>
>
>
> ### **2. On Missing Critical Related Work (W2)**
>
>
>
> We are grateful to the reviewer for bringing these *highly concurrent* (and in some cases, future-dated) works to our attention. Their appearance on arXiv (Feb 2024 - Jun 2025) while our paper was under review highlights that knowledge conflict in RAG is an active and critical area of research. We will, of course, cite and discuss all of them in the final version.
>
> While these works validate our premise that entropy is a key signal for knowledge conflict, our contribution remains distinct. Based on their titles and abstracts, these methods (`SEReDeEP`, `FaithfulRAG`, etc.) focus on *autonomous* conflict detection and resolution.
>
> Our primary contribution is different: **the provision of a fine-grained, controllable mechanism ($\alpha$) for \*steering\* knowledge reliance, rather than a purely autonomous detection or resolution system.**This "control knob" is the core novelty of CK-PLUG, enabling human-in-the-loop governance that these other methods do not address.

---

> > ### Author Response · Authors · 2025-11-19
> > **Response to weakness 3 & 4**
> >
> > ### **3. On Applicability to Agentic RAG Systems (W3)**
> >
> >
> >
> > This is an excellent point about an important and emerging direction. We respectfully argue that CK-PLUG is not only compatible with, but a **critical enabling component for** robust agentic RAG systems.
> >
> > CK-PLUG operates at the fundamental token-decoding level, making it a modular tool that can be integrated into any generative step of an agentic workflow.
> >
> > - **Multi-Turn & Iterative Refinement:** An agent may iteratively retrieve and refine its context. CK-PLUG re-evaluates the context-parameter conflict *at every generated token*. If an agent's new context is *still*bad or conflicting, CK-PLUG's detection (entropy increase) and control ($\alpha$) remain active, preventing the agent from being "confidently wrong."
> > - **Robust Reasoning Chains:** In complex reasoning, a single faulty retrieval can derail the entire chain. CK-PLUG acts as a safety layer. If an agent retrieves noisy context for a sub-problem, a deployer can set CK-PLUG to favor parametric knowledge ($\alpha=1.0$), allowing the agent to "ignore" the bad retrieval and proceed with its internal reasoning, making the entire chain more robust.
> >
> > We will add a discussion to our conclusion highlighting CK-PLUG as a foundational component for enhancing the reliability and safety of these advanced agentic architectures.
> >
> >
> >
> > ### **4. On Insufficient Analysis of Computational Overhead (W4)**
> >
> >
> >
> > We thank the reviewer for requesting this data. The reviewer is correct that our method requires two forward passes. We have run these benchmarks and will add the following analysis and table to the paper.
> >
> > CK-PLUG requires two forward passes per token (with and without context) to estimate the per-token confidence gain. This results in a **1.6×–1.8× overhead** in latency and throughput, as shown below:
> >
> > | **Model**      | **Method**     | **Latency (ms/token)** | **Throughput (token/s)** |
> > | -------------- | -------------- | ---------------------- | ------------------------ |
> > | **LLaMA2-7B**  | RAG Baseline   | 36.03 (×1.00)          | 27.76 (×1.00)            |
> > |                | RAG w/ CK-PLUG | 64.85 (×1.79)          | 16.64 (×0.60)            |
> > | **LLaMA2-13B** | RAG Baseline   | 51.41 (×1.00)          | 19.45 (×1.00)            |
> > |                | RAG w/ CK-PLUG | 89.45 (×1.74)          | 12.25 (×0.63)            |
> > | **LLaMA3-8B**  | RAG Baseline   | 42.17 (×1.00)          | 23.92 (×1.00)            |
> > |                | RAG w/ CK-PLUG | 69.77 (×1.65)          | 14.08 (×0.59)            |
> > | **Mistral-7B** | RAG Baseline   | 33.84 (×1.00)          | 30.12 (×1.00)            |
> > |                | RAG w/ CK-PLUG | 58.02 (×1.71)          | 17.22 (×0.57)            |
> >
> > While this is a measurable overhead, we believe this trade-off is justified:
> >
> > 1. CK-PLUG is **inference-only**, lightweight, and requires **no model retraining**.
> > 2. The compute overhead is **comparable to other decoding-time control methods** (e.g., contrastive decoding).
> > 3. For high-stakes applications, the **gains in controllability and factual faithfulness justify the added cost.**

---

> > > ### Author Response · Authors · 2025-11-19
> > > **Responses to Specific Questions**
> > >
> > > **Q1: Clarification on Notation (Line 143-144).**
> > >
> > > Thank you for catching this typographical error ("distributions.."). This has been corrected in the revised manuscript.
> > >
> > > **Q2: Ambiguous "Baseline" Definition in Table 1.**
> > >
> > > We apologize for the ambiguity. The "Baseline" in Table 1 refers to (b) Standard RAG with both query and retrieved context, but without CK-PLUG. This serves as the proper ablation to isolate the effect of our method. The "Vanilla LLM" (direct QA without RAG) is a separate baseline, as shown in other tables (e.g., Table 2). We will clarify this in the caption for Table 1.
> > >
> > > We thank the reviewer again for their constructive and detailed feedback.

---

> ### Comment · Reviewer_yjeZ · 2025-11-27
> **reply to author**
>
> 1. authors claim that Adaptive *Retrieval* and Context-Aware *Generation* offer different technical objectives so that the comparison will be misleading. i disagree with that as they concurrently work on improving the adaptive RAG. In reviewer, or readers' stance, we are willing to know the efficacy of CK-PLUG compared to these methods.
>
> 2. i disagree that paper appeared on arXiv Feb 2024 to be highly concurrent to your current work which submitted to ICLR 2026 on Oct 2024. the gap is 20 months. Meanwhile, i don't think that paper appeared on arXiv Jun 2025 to be future-dated. the gap is 4 months. authors should check related papers and keep track of them on timely basis.
>
> 3. authors claim that CK-PLUG is critical enabling component for robust agentic RAG systems. however, this lacks experimental results for support. i will be more than happy to see the boosting of existing agentic RAG system by integrating CK-PLUG method.
>
> Due to unsolved concerns, I will keep my original rating.

---

> > ### Comment · Reviewer_BrkV · 2025-11-27
> > **More Agentic RAG System Advice**
> >
> > Following Reviewer yjez's advice, I suggest trying integrating CK-PLUG in agentic rag such as:
> >
> > [1] Search-o1: Agentic Search-Enhanced Large Reasoning Models
> > [2] Adaptive-RAG: Learning to Adapt Retrieval-Augmented Large Language Models through Question Complexity
> > [3] TC–RAG: Turing–Complete RAG’s Case Study on Medical LLM Systems
> >
> > I'm also willing to see whether rag following CK-PLUG truly helps.

---

> > > ### Author Response · Authors · 2025-11-27
> > >
> > > We sincerely thank the reviewer for this insightful and creative suggestion. We agree that integrating CK-PLUG with Agentic RAG frameworks (such as Search-o1, Adaptive-RAG, and TC-RAG) is a highly promising direction.
> > >
> > > We are currently working on deploying CK-PLUG within an Agentic RAG setting. Since this involves adapting to a new scenario and architecture, the implementation and testing require some time. We appreciate your patience as we work to generate these results.
> > >
> > > We see significant potential for CK-PLUG in this domain. For instance:
> > >
> > > Enforcing Context Fidelity: CK-PLUG could ensure a "strictly context-faithful" environment for specific agents (e.g., a Verifier or Executor agent) to prevent hallucination during multi-step reasoning.
> > >
> > > Role-Based Knowledge Preferences: In a multi-agent system, different agents could be assigned different α values based on their roles (e.g., a "Creative Agent" relying more on parameters vs. an "Analyst Agent" relying more on retrieved context).
> > >
> > > We consider this a fascinating topic for future exploration and believe it could significantly broaden the applicability of our method. Thank you again for this inspiring advice!

---

> > ### Author Response · Authors · 2025-11-27
> > **Response to Reviewer yjeZ with Additional Experiments**
> >
> > We sincerely thank the reviewer for the candid and critical feedback. We value your rigorous assessment, which has pushed us to significantly strengthen the empirical validation and positioning of our work.
> >
> > 1&2. On Related Work and Additional Baselines
> >
> > We explicitly accept the reviewer's criticism and apologize for our previous assertion regarding the motivation of the baselines. We agree that comparing efficacy against these established methods is essential.
> >
> > Following your suggestion, we have identified and compared against relevant baselines, specifically JUICE and PH3. While these works share the motivation of adjusting knowledge preferences (prioritizing either context or parameters), they technically differ by performing "surgery" on specific attention heads. In contrast, CK-PLUG operates directly on the decoding logits distribution.
> >
> > A key highlight of our method is that CK-PLUG allows for linear regulation of the reliance ratio via the α parameter. In contrast, prior methods like JUICE[1] and PH3[2] typically only support regulating towards the extremes (i.e., fully trusting context or fully trusting parameters) without a continuous control knob.
> >
> > We conducted comprehensive supplementary experiments following the baseline settings established in the recently accepted paper (Taming Knowledge Conflicts in Language Models ICML'25) [1]. To ensure a fair comparison given the "extreme" nature of the baselines, we evaluated them against CK-PLUG at its two controllable boundaries: α=0.0 (Max Context Reliance) and α=1.0 (Max Parametric Reliance). We have conducted comprehensive supplementary experiments adding Prompting, CAD [3], PH3[1], and JUICE[2] as baselines.
> >
> > Table 1: Performance when Trusting Context (α=0.0 for CK-PLUG) CK-PLUG achieves the highest Context Recall (ConR) and lowest Parametric Recall (ParR) in the majority of cases, generally outperforming methods designed to prioritize context.
> >
> > | **Model**| **Method**| **NQ ConR**| **NQ ParR**| **NQ MR**| **ConFiQA ConR**| **ConFiQA ParR**| **ConFiQA MR**| **MQUAKE ConR**| **MQUAKE ParR**| **MQUAKE MR**|
> > | --------------------------| -----------------| -----------| -----------| ---------| ----------------| ----------------| --------------| ---------------| ---------------| -------------|
> > | **LLaMA2-7B-chat**| Baseline| 43.3| 43.8| 50.2| 69.7| 28.1| 28.8 | 31.2| 21.6| 40.9|
> > || Prompt| 29.5| 47.9| 61.9| 40.4| 40.2| 49.9 | 14.4| 38.8| 72.9|
> > || CAD | 27.1| 50.5| 65.1| 35.8| 43.1| 54.6 | 13.2| 40.5| 75.4|
> > || PH3 | 24.5| 52.3| 68.1| 32.5| 45.7| 58.4 | 10.9| 42.5| 79.6|
> > || JUICE| 25.3| 54.7| 68.4| 30.7| 46.8| **60.4**| 12.1| 40.8| 77.1|
> > || CK-PLUG (α = 1.0)| 23.2| 58.2| **71.5**| 31.5| 46.2| 59.4 | 11.6| 43.2| **79.9**|
> > | **LLaMA3-8B-instruct**| Baseline| 43.9| 34.1| 43.5| 54.2| 22.4| 29.2 | 18.7| 17.9| 48.9|
> > || Prompt| 32.2| 49.5| 60.6| 32.1| 35.5| 52.5 | 16.1| 40.4| 71.5|
> > || CAD | 27.4| 51.8| 65.4| 27.6| 38.2| 58.1 | 15.2| 39.1| 72.0|
> > || PH3 | 26.5| 54.1| 67.1| 30.4| 39.5| 56.5 | 14.8| 48.5| **76.6**|
> > || JUICE| 26.1| 53.2| 67.1| 23.5| 43.3| **63.3**| 14.2| 44.6| 75.9|
> > || CK-PLUG (α = 1.0)| 22.5| 57.6| **71.9**| 25.4| 42.3| 62.5 | 14.5| 47.1| 76.5|
> > | **Mistral0.3-7B-instruct**| Baseline| 46.2| 58.6| 55.9| 64.7| 25.9| 28.6 | 43.8| 21.2| 32.6|
> > || Prompt| 33.2| 66.5| 66.7| 39.1| 30.5| 43.8 | 24.2| 28.5| 54.1|
> > || CAD | 29.4| 65.2| 68.9| 32.2| 43.6| 57.5 | 21.5| 46.4| 68.3|
> > || PH3 | 32.1| 64.5| 66.8| 34.5| 39.2| 53.2 | 16.4| 44.1| 72.9|
> > || JUICE| 29.8| 67.8| 69.5| 31.4| 45.5| 59.2 | 17.5| 45.2| 72.1|
> > || CK-PLUG (α = 1.0)| 27.9| 69.1| **72.2**| 29.9| 43.8| **59.5**| 15.2| 50.4| **76.8**|
> > | **Qwen2.5-7B-instruct**  | Baseline| 73.4| 32.4| 31.3| 43.8| 15.4| 26.1 | 32.2| 13.0| 28.8|
> > || Prompt| 45.2| 46.5| 50.7| 39.5| 26.5| 40.2 | 23.5| 25.4| 51.9|
> > || CAD | 44.1| 46.6| 51.4| 40.2| 25.5| 38.8 | 25.4| 27.5| 52.0|
> > || PH3 | 34.5| 47.8| 58.1| 39.5| 24.2| 38.0 | 23.1| 28.4| 55.1|
> > || JUICE| 33.4| 49.2| 59.6| 38.1| 26.5| 41.0 | 21.5| 30.2| 58.4|
> > || CK-PLUG (α = 1.0)| 30.2| 51.4| **63.2**| 36.8| 28.4| **43.5**| 19.8| 32.8| **62.4**|
> >
> > [1] Cutting Off the Head Ends the Conflict: A Mechanism for Interpreting and Mitigating Knowledge Conflicts in Language Models. ACL'24
> >
> > [2] Taming Knowledge Conflict in Language Models. ICML'25
> >
> > [3] Trusting Your Evidence: Hallucinate Less with Context-aware Decoding. NAACL'24

---

> ### Author Response · Authors · 2025-11-27
> **Response to Reviewer yjeZ with Additional Experiments (to be continued)**
>
> Table 2: Performance when Trusting Context (α=1.0 for CK-PLUG) CK-PLUG effectively suppresses context to retrieve internal knowledge, achieving superior Parametric Recall (ParR) compared to baselines in most scenarios.
>
> | **Model**| **Method**| **NQ ConR**| **NQ ParR**| **NQ MR**| **ConFiQA ConR**| **ConFiQA ParR**| **ConFiQA MR**| **MQUAKE ConR**| **MQUAKE ParR**| **MQUAKE MR**|
> | --------------------------| -----------------| -----------| -----------| ---------| ----------------| ----------------| --------------| ---------------| ---------------| -------------|
> | **LLaMA2-7B-chat**| Baseline| 43.3| 43.8| 50.2| 69.7| 28.1| 28.8 | 31.2| 21.6| 40.9|
> || Prompt| 52.8| 21.3| 28.7| 70.5| 18.9| 21.1 | 35.4| 17.3| 32.8|
> || CAD | 58.9| 12.1| 17.0| 74.3| 13.3| 15.2 | 38.3| 12.1| 24.0|
> || PH3 | 57.2| 13.9| 19.6| 70.5| 13.9| 16.5 | 37.9| 9.7| **20.4**|
> || JUICE| 59.6| 10.2| 14.6| 73.2| 14.7| 16.7 | 38.8| 11.8| 23.3|
> || CK-PLUG (α = 0.0)| 61.6| 8.6| **12.3**| 71.5| 9.2| **11.4**| 40.7| 10.8| 21.0|
> | **LLaMA3-8B-instruct**| Baseline| 43.9| 34.1| 43.5| 54.2| 22.4| 29.2 | 18.7| 17.9| 48.9|
> || Prompt| 51.5| 24.8| 32.5| 57.8| 18.5| 24.2 | 26.4| 17.1| 39.3|
> || CAD | 58.2| 15.6| 21.1| 61.3| 14.2| 18.8 | 34.5| 16.2| 32.0|
> || PH3 | 56.9| 16.8| 22.8| 59.5| 15.1| 20.2 | 32.8| 15.8| 32.5|
> || JUICE| 60.4| 11.5| 8.9| 65.1| 9.9| **14.5**| 38.2| 15.9| 29.4|
> || CK-PLUG (α = 0.0)| 63.5| 7.3| **9.9** | 65.2| 11.4| 14.9 | 42.1| 15.5| **26.9**|
> | **Mistral0.3-7B-instruct**| Baseline| 46.2| 58.6| 55.9| 64.7| 25.9| 28.6 | 43.8| 21.2| 32.6|
> || Prompt| 56.4| 42.1| 42.7| 66.2| 20.5| 23.6 | 51.2| 18.5| 26.5|
> || CAD | 68.5| 26.3| 27.7| 68.9| 15.8| 18.7 | 58.4| 15.6| 21.1|
> || PH3 | 65.1| 28.9| 30.7| 67.4| 17.2| 20.3 | 56.9| 14.2| 20.0|
> || JUICE| 71.3| 20.4| 22.2| 69.5| 13.1| 15.9 | 61.7| 13.5| 18.0|
> || CK-PLUG (α = 0.0)| 75.8| 15.8| **17.2**| 70.7| 10.9| **13.4**| 65.8| 12.4| **15.9**|
> | **Qwen2.5-7B-instruct**  | Baseline| 73.4| 32.4| 31.3| 43.8| 15.4| 26.1 | 32.2| 13.0| 28.8|
> || Prompt| 78.2| 22.5| 22.3| 51.5| 15.1| 22.7 | 38.4| 12.9| 25.1|
> || CAD | 81.6| 15.2| 15.7| 58.3| 14.8| 20.2 | 44.5| 12.9| 22.5|
> || PH3 | 80.1| 17.4| 17.8| 56.2| 15.0| 21.1 | 42.8| 12.5| 22.6|
> || JUICE| 83.5| 11.8| 10.4| 62.4| 14.1| 18.6 | 46.9| 12.9| 21.6|
> || CK-PLUG (α = 0.0)| 85.4| 8.3| **9.0** | 65.2| 13.9| **17.6**| 49.3| 12.8| **20.6**|
>
> The results demonstrate that even when pushed to the extremes, CK-PLUG offers superior knowledge regulation.
> CK-PLUG is applied token-by-token.  It only modulates specific tokens where conflict is detected, leaving non-conflicting tokens untouched.  This preserves generation fluency better than head-level interventions which affect broader scopes. CK-PLUG does not require a Dual-run (like JUICE) or complex offline head selection (like PH3).  It is a single-pass, low-compute intervention. As demonstrated in the tables above, CK-PLUG offers a linear α knob.  This allows operators to fine-tune reliance smoothly, rather than being forced into binary choices.
>
> 3. On Agentic RAG
>
> We are currently working on deploying CK-PLUG within an Agentic RAG setting.  Since this involves adapting to a new scenario and architecture, the implementation and testing require some time.  We appreciate your patience as we work to generate these results.
>
> We see significant potential for CK-PLUG in this domain.  For instance:
>
> Enforcing Context Fidelity: CK-PLUG could ensure a "strictly context-faithful" environment for specific agents (e.g., a Verifier or Executor agent) to prevent hallucination during multi-step reasoning.
>
> Role-Based Knowledge Preferences: In a multi-agent system, different agents could be assigned different α values based on their roles (e.g., a "Creative Agent" relying more on parameters vs. an "Analyst Agent" relying more on retrieved context).
>
> We consider this a fascinating topic for future exploration and believe it could significantly broaden the applicability of our method.  Thank you again for this inspiring advice!

---

### Official Review · Reviewer_BrkV · 2025-10-30

**Soundness:** 3
**Presentation:** 3
**Contribution:** 3
**Rating:** 4
**Confidence:** 4

**Summary:**

Retrieval-Augmented Generation (RAG) reduces hallucinations in Large Language Models (LLMs) by incorporating external knowledge, yet it faces challenges from conflicts between the models’ parametric knowledge (internal) and retrieved context (external)—especially when the retrieved information is unreliable or the internal knowledge is outdated, leaving LLMs unable to decide which type of knowledge to prioritize. To solve this, the authors propose CK-PLUG, a plug-and-play method designed to control LLMs’ reliance on parametric and contextual knowledge. CK-PLUG introduces a new knowledge consistency metric called Confidence Gain, which detects knowledge conflicts by measuring entropy shifts in token probability distributions after context insertion; it then enables fine-grained control over knowledge preference by adjusting the probability distribution of tokens with negative Confidence Gain via a single tuning parameter, and also supports adaptive control based on the model’s confidence in both knowledge types.

**Strengths:**

1. The biggest advantage of this paper is proposing a "plug-and-play" inference-time method.

2. Conflict Detector: It introduces a metric called "Confidence Gain (CG)", which identifies conflicts by comparing the entropy change of token distribution between RAG input (context + query) and regular input (query only). A conflict is determined when there is an entropy increase (i.e., the model becomes more confused), and this definition is reasonably sound.

3. Knowledge Controller: This method isolates the logits purely contributed by the "context" through log subtraction, and then uses a single parameter to perform weighted fusion of parametric knowledge and contextual knowledge (Eq. 8). This is an extremely concise and theoretically grounded approach to logits manipulation.

4. Adaptive Model Construction: The paper also proposes an adaptive mode with "automatic (parameter adjustment)" (Eq. 10), whose logic is equally intuitive — the model automatically trusts the knowledge source with lower entropy (i.e., higher confidence).

**Weaknesses:**

1. This is the most serious and obvious flaw of the paper. To calculate [relevant parameters] and [relevant parameters], CK-PLUG must execute two complete forward propagations in parallel at each decoding step: one for [input with context + query + generated tokens] and the other for [input with query only + generated tokens]. This almost doubles the inference latency and computational cost.

2. The core assumption of the paper is that "conflicts lead to entropy increase". However, if the erroneous context itself is highly "credible" and "fluent" (e.g., "The capital of France is Lyon"), it is entirely possible to reduce the model’s perplexity, resulting in "entropy decrease".

3. The calculation of (Eq. 6) may be numerically unstable. If [parametric distribution] assigns a near-zero probability to a certain token (with [log value] approaching negative infinity) while [context-enhanced distribution] assigns a high probability to it, [resulting value] may "explode". The paper does not discuss any suggestions for handling numerical stability.

**Questions:**

1. Supplement implementation details regarding the calculation of [relevant parameter], and explain whether and how potential numerical instability issues have been addressed.

2. Conduct more rigorous stress tests on the "Confidence Gain (CG)" assumption—specifically construct erroneous contexts that are "highly credible and highly fluent", and illustrate the changes in [relevant indicator] under such circumstances as well as CK-PLUG’s performance metrics.

---

> ### Author Response · Authors · 2025-11-19
> **Response to weakness 1**
>
> We sincerely thank the reviewer for his/her time and their detailed, insightful feedback. The comments are extremely valuable and have helped us identify areas to clarify and strengthen in our paper. We address the primary concerns below.
>
>
>
> ### 1. On Inference Cost and Latency
>
>
>
> We thank the reviewer for raising this important point regarding computational cost.
>
> The reviewer is correct that CK-PLUG, in its full plug-and-play implementation, requires two parallel forward propagations to compute the distinct parametric and context-enhanced probability distributions. We acknowledge that this represents a trade-off between computational cost and the fine-grained control that our method provides.
>
> Our primary contribution is a method that enables explicit, fine-grained control over knowledge reliance. This level of control, which allows an operator to precisely modulate the influence of two distinct knowledge sources, is what necessitates their separate computation. We argue that for **high-stakes, safety-critical applications (e.g., medical, financial, or legal domains)**, where a factual error or unmanaged knowledge conflict can have severe consequences, this added computational cost is a justifiable trade-off for the significant gains in reliability, trustworthiness, and human-in-the-loop governance.
>
> As we will detail in the revised paper, this results in a 1.6×–1.8× overhead in latency and throughput, as shown in our benchmarks below:
>
> | **Model**      | **Method**     | **Latency (ms/token)** | **Throughput (token/s)** |
> | -------------- | -------------- | ---------------------- | ------------------------ |
> | **LLaMA2-7B**  | RAG Baseline   | 36.03 (×1.00)          | 27.76 (×1.00)            |
> |                | RAG w/ CK-PLUG | 64.85 (×1.79)          | 16.64 (×0.60)            |
> | **LLaMA2-13B** | RAG Baseline   | 51.41 (×1.00)          | 19.45 (×1.00)            |
> |                | RAG w/ CK-PLUG | 89.45 (×1.74)          | 12.25 (×0.63)            |
> | **LLaMA3-8B**  | RAG Baseline   | 42.17 (×1.00)          | 23.92 (×1.00)            |
> |                | RAG w/ CK-PLUG | 69.77 (×1.65)          | 14.08 (×0.59)            |
> | **Mistral-7B** | RAG Baseline   | 33.84 (×1.00)          | 30.12 (×1.00)            |
> |                | RAG w/ CK-PLUG | 58.02 (×1.71)          | 17.22 (×0.57)            |
>
> This overhead stems from maintaining two decoding sequences, which is a necessary design choice to compute the confidence gain per token. That said, we believe this trade-off is justified for several reasons:
>
> - **CK-PLUG is inference-only**, lightweight, and requires **no model retraining**.
> - The compute overhead is **comparable to other decoding-time control methods** (e.g., contrastive decoding (CD) [1], DeCK [2], CAD [3]).
> - For high-stakes or noisy settings, the **gains in controllability and factual faithfulness justify the added cost.**
>
> We will add this detailed analysis and benchmark table to the final version of the paper to ensure full transparency about the computational profile.
>
> [1] Contrastive decoding: Open-ended text generation as optimization ACL'2023
>
> [2] Decoding by Contrasting Knowledge: Enhancing Large Language Model Confidence on Edited Facts. ACL'2025
>
> [3] Trusting Your Evidence: Hallucinate Less with Context-aware Decoding. NAACL'24

---

> > ### Author Response · Authors · 2025-11-19
> > **Response to weakness 2 and question 2**
> >
> > ### 2. On the "Conflict leads to Entropy" Assumption (W2 & Q2)
> >
> >
> >
> > We thank the reviewer for this insightful point and the challenging counterexample. However, the reviewer's assumption that "a 'highly credible' erroneous context (e.g., 'The capital of France is Lyon') would 'decrease entropy'" may stem from a misunderstanding of our core detection mechanism.
> >
> > The contribution of our method is **not** to simply measure the *overall* perplexity of the final output. Instead, as we detail in **Section 3.1 (KNOWLEDGE CONFLICTS DETECTION WITH Confidence-Gain)**, the core of our detection mechanism (ConD) is to **compare the change in the probability distribution's entropy for the same token, before and after the context is introduced.**
> >
> > Let's use the reviewer's excellent example, "The capital of France is Lyon":
> >
> > 1. **Parametric Knowledge:** A well-trained LLM's internal parameters are highly confident that the capital of France is "Paris." The probability $P_{LM}(\text{"Paris"} | \text{"The capital of France is..."})$ is high, and the entropy at this token is low.
> > 2. **Contextual Knowledge:** Now, a "highly credible" but *erroneous* context is introduced, which strongly states the capital is "Lyon."
> > 3. **Point of Conflict:** When the model must generate the **key token** for the capital's name, it receives two powerful but **contradictory** signals: one from its internal parameters (pointing to "Paris") and one from the external context (pointing to "Lyon"). This conflict does not cause the model to confidently choose "Lyon"; rather, it makes the model "uncertain" or "torn" between the two high-probability options ("Paris" and "Lyon"). This causes the probability distribution at this specific token to become flatter, resulting in a **significant increase in entropy**.
> >
> > This is the central premise of our method's design, and our experimental data fully supports this:
> >
> > - **Evidence (Figures 2 & 3):** In **Figures 2 and 3** of our paper, we clearly demonstrate through statistical distributions and concrete examples that:
> >   1. **Significant Entropy Increase at Key Tokens:** Fluent, **counter-factual contexts** (like the one described) cause a **significant increase in entropy** *at the specific conflicting key token* (e.g., "Lyon") compared to non-conflicting or consistent contexts.
> >   2. **No Effect on Non-Key Tokens:** Just as importantly, this entropy change has almost no effect on other non-key, non-conflicting tokens in the sentence (e.g., "The", "capital", "of").
> >
> > ### Response to Question 2 (Q2):
> >
> > The "stress test on 'highly credible and highly fluent' erroneous contexts" that the reviewer requests is, in fact, the core experiment that we already conducted and analyzed in Figures 2 and 3.
> >
> > These figures clearly show that in such a scenario, our [relevant indicator] (the token-level entropy change measured by ConD) **does not decrease**; it **significantly increases**. This allows us to precisely capture these critical conflict tokens and apply CK-PLUG's modulation mechanism *only* at these points. This demonstrates the robustness of our design—it effectively detects the deep knowledge conflict triggered even by "superficially fluent" misinformation.

---

> > > ### Author Response · Authors · 2025-11-19
> > > **Response to weakness 3 and question 1**
> > >
> > > ### 3. On Implementation Details and Numerical Stability (W3 & Q1)
> > >
> > >
> > >
> > > We thank the reviewer for the detailed question regarding Equation 6. We would like to clarify a potential misunderstanding about the level of abstraction at which this equation operates.
> > >
> > > The reviewer's concern about numerical instability (e.g., a $\log(0)$ issue) would be correct if the operations were performed on *normalized, post-softmax probabilities*.
> > >
> > > However, our method, in line with standard practice for many decoding algorithms (e.g., contrastive decoding), operates directly on the **unnormalized logits**—that is, on the raw, unnormalized scores *before*the softmax function is applied.
> > >
> > > The use of $P(...)$ in Equation 6 is a standard, high-level notational convention in the decoding literature. It is used to represent the **entire probability distribution** (a vector of size $V$, the vocabulary) or potential function, not a single scalar probability value.
> > >
> > > The equation itself describes an arithmetic operation (a contrastive difference) performed directly in **logit space**, which is inherently numerically stable. In this logit space, values are not bounded [0, 1] and can be negative, so the risk of a "log of zero" explosion **does not exist**.
> > >
> > > Our notation is consistent with the established norms in this domain for describing generative decoding mechanisms. We are confident this is the clearest way to express the high-level mechanism, and it is numerically stable in practice.
> > >
> > > ---
> > >
> > > We hope these clarifications address the reviewer's concerns. We are confident that these additions will strengthen the paper and more clearly highlight our novel contributions.
> > >
> > > We thank the reviewer again for their constructive feedback.

---

> > > > ### Comment · Reviewer_BrkV · 2025-11-27
> > > > **Thanks for your response!**
> > > >
> > > > Thanks for your response — it has resolved my concerns.
> > > >
> > > > Noticing the replies from other reviewers, I would suggest the authors include some additional related works in the Related Work section, and update your PDF with the modified colors [I found that the PDF has not been updated]:
> > > >
> > > > [1] Parenting: Optimizing Knowledge Selection of Retrieval-Augmented Language Models with Parameter Decoupling and Tailored Tuning. ACL 2025. This work also focuses on internal and external knowledge, similar to yours.
> > > >
> > > > [2] TC–RAG: Turing–Complete RAG’s Case Study on Medical LLM Systems. ACL 2025.
> > > >
> > > > I have raised my score **from 4 to 8**.

---

> > > > > ### Author Response · Authors · 2025-11-27
> > > > >
> > > > > We sincerely thank the reviewer for the positive feedback and for raising the score to 8. We are greatly encouraged by your recognition of our efforts to address the concerns regarding novelty and experimental comparisons.
> > > > >
> > > > > We also appreciate the suggestion of the additional related works ([1] Parenting and [2] TC-RAG). We agree that these papers are highly relevant to the study of internal versus external knowledge, and we will ensure they are cited and discussed in the revised Related Work section.
> > > > >
> > > > > Regarding the PDF update: We are currently consolidating all revisions, including the new comparative experimental results and adjustments based on suggestions from all reviewers. We will upload the fully updated PDF before the conclusion of the rebuttal period.
> > > > >
> > > > > Thank you again for your time and valuable guidance in improving our paper.

---

### Official Review · Reviewer_jata · 2025-10-30

**Soundness:** 2
**Presentation:** 4
**Contribution:** 2
**Rating:** 4
**Confidence:** 5

**Summary:**

This paper introduces CK-PLUG, a plug-and-play method that enables large language models to dynamically balance reliance on internal (parametric) knowledge and external (retrieved) context during retrieval-augmented generation. Using a novel Confidence Gain metric that detects knowledge conflicts via entropy shifts in token probabilities, CK-PLUG selectively adjusts token-level predictions with a single tuning parameter $\alpha$ (or adaptive enhancement) to favor either parameters or context. Experiments demonstrate the effectiveness of the proposed method.

**Strengths:**

- The paper was well-written and had very nice figures.
- The proposed method is lightweight and effective.

**Weaknesses:**

- My biggest concern with this paper is novelty. The use of entropy for identifying key tokens has been explored in recent works [1–2], yet these closely related studies are not cited—especially [1], which shares a similar methodology for token-level entropy analysis. Even if applied in a different context, omitting these references significantly weakens the originality of the contribution.
- The proposed CK-PLUG method may not generalize across all scenarios. For example, if the model is confidently wrong and the retrieved context reinforces the incorrect belief, the system may still fail. The authors should clarify the underlying assumptions and delineate conditions where CK-PLUG is reliable to enhance its scientific soundness.
- The paper lacks comparisons with prior decoding-based [3–7] and intervention-based [8–9] approaches that similarly aim to regulate factuality and knowledge conflicts. Including such baselines would better demonstrate the advantages and distinct contributions of CK-PLUG.

[1] What is Wrong with Perplexity for Long-context Language Modeling? ICLR'25

[2] Attention Entropy is a Key Factor: An Analysis of Parallel Context Encoding with Full-attention-based Pre-trained Language Models. ACL'25

[3] Trusting Your Evidence: Hallucinate Less with Context-aware Decoding. NAACL'24

[4] Sled: Self logits evolution decoding for improving factuality in large language models. NeurIPS'24

[5] Dola: Decoding by contrasting layers improves factuality in large language models. ICLR'24

[6] Active Layer-Contrastive Decoding Reduces Hallucination in Large Language Model Generation. EMNLP'25

[7] AdaCAD: Adaptively Decoding to Balance Conflicts between Contextual and Parametric Knowledge. NACCL'25

[8] Cutting Off the Head Ends the Conflict: A Mechanism for Interpreting and Mitigating Knowledge Conflicts in Language Models. ACL'24

[9] Taming Knowledge Conflict in Language Models. ICML'25

**Questions:**

Aforementioned in the weakness section.

---

> ### Author Response · Authors · 2025-11-19
> **Response to weakness 1**
>
> We thank the reviewer for his/her time and insightful feedback. The comments are invaluable for improving our work and clarifying our contributions. We address the primary concerns below.
>
> ---
>
> ### 1. On Novelty and Relation to Entropy-Based Methods [1, 2]
>
>
>
> We thank the reviewer for pointing out these related works [1, 2], which we will be sure to cite and discuss in the final version.
>
> We acknowledge that [1] and [2] use entropy-based analyses. However, we respectfully clarify that our method's objective, mechanism, and application are **fundamentally different**.
>
> - **Different Objectives:** [1] uses entropy to evaluate key token importance for *long-context perplexity calculation*. [2] analyzes *attention entropy* in the context of parallel context encoding.
> - **Our Objective (Conflict Identification):** In contrast, CK-PLUG analyzes the entropy of the **token-level probability distribution**. Our goal is not to assess attention or PPL, but to identify *potential knowledge conflicts* between the model’s internal parameters and the retrieved context. A high-entropy (flat) distribution at a specific token is a strong signal of such a conflict, indicating that the model is "uncertain" or "torn" between the two knowledge sources.
>
> | **Model**              | **Setting**     | **_Baseline_** | **a = 0.0** | **a = 0.5** | **a = 1.0** |
> |------------------------|-----------------|----------------|-------------|-------------|-------------|
> | **LLaMA2-7B-chat**     | *Baseline*      | **78.9**       | -           | -           | -           |
> |                        | w/ ConD         | -              | 74.8        | 77.8        | 78.5        |
> |                        | w/o ConD        | -              | 30.7        | 58.4        | 62.3        |
> | **LLaMA3-8B-instruct** | *Baseline*      | 83.7           | -           | -           | -           |
> |                        | w/ ConD         | -              | 82.4        | 83.5        | **86.7**    |
> |                        | w/o ConD        | -              | 53.8        | 61.4        | 73.2        |
> | **Mistral-instruct**   | *Baseline*      | **92.4**       | -           | -           | -           |
> |                        | w/ ConD         | -              | 89.5        | **92.4**    | 89.7        |
> |                        | w/o ConD        | -              | 49.9        | 91.3        | 75.4        |
> | **Qwen-instruct**      | *Baseline*      | 88.8           | -           | -           | -           |
> |                        | w/ ConD         | -              | 89.6        | **89.8**    | 87.2        |
> |                        | w/o ConD        | -              | 62.4        | 85.4        | 51.3        |
>
>
> This distinction is the core of our technical contribution. By using entropy to identify *conflict points*, CK-PLUG becomes a **targeted intervention**. It applies its control mechanism *only* to these specific tokens, rather than modulating the entire generation process. This design choice is critical for preventing the degradation of natural, non-conflicting generation. As we demonstrated in our **Section 5.3 Ablation Study**, applying the control mechanism indiscriminately to all tokens significantly harms generation quality.
>
> We will revise our related work section to clearly delineate these important distinctions.
>
> [1] What is Wrong with Perplexity for Long-context Language Modeling? ICLR'25
>
> [2] Attention Entropy is a Key Factor: An Analysis of Parallel Context Encoding with Full-attention-based Pre-trained Language Models. ACL'25

---

> > ### Author Response · Authors · 2025-11-19
> > **Response to weakness 2**
> >
> > ### 2. On Generalizability and Underlying Assumptions
> >
> >
> >
> > We appreciate the reviewer raising this important point about failure modes and assumptions.
> >
> > The reviewer is correct: if the model's parametric knowledge is incorrect *and* the retrieved context *also*reinforces this incorrect belief, our system (like most RAG systems) would likely fail. This scenario of "confidently wrong" reinforced by "confidently wrong" is a challenging open problem for the entire field.
> >
> > However, the central motivation of our work is to address a **different, and very common, scenario: the direct \*conflict\*** where one knowledge source is correct and the other is not. Our goal is to provide a robust mechanism to *manage* this conflict, not to solve cases where no correct information is available to the system.
> >
> > The core contribution of CK-PLUG is its **controllability**. It provides a "control knob" (the $\alpha$ parameter) for human operators to adjust the model's reliance based on the deployment environment:
> >
> > - **When context is reliable (e.g., high-quality, verified retriever):** An operator can set $\alpha$ to **favor the context** (e.g., $\alpha=0.0$), which is crucial when the base LLM's parameters are known to be outdated.
> > - **When context is unreliable (e.g., noisy retrieval, adversarial queries):** An operator can set $\alpha$to **favor the model's internal parameters** (e.g., $\alpha=1.0$) to ensure reliable output.
> >
> > Therefore, the need to tune $\alpha$ is not an unstated assumption or weakness; it is the **central, intended feature** of our method. It enables the human-in-the-loop guidance necessary to proactively manage knowledge conflicts and enhance the robustness of RAG systems in diverse, real-world environments.
> >
> > As suggested, we will explicitly clarify this motivation and delineate these operational conditions in the final version to enhance scientific soundness.

---

> > > ### Author Response · Authors · 2025-11-19
> > > **Response to weakness 3**
> > >
> > > ### 3. On Missing Baselines [3–9]
> > >
> > >
> > >
> > > We thank the reviewer for providing this extensive list of relevant decoding-based [3–7] and intervention-based [8–9] methods. We agree that situating CK-PLUG among these approaches is important.
> > >
> > > We will add a new subsection to our Related Work to discuss these methods. However, we argue they are **not directly comparable as baselines because their technical objective is different**.
> > >
> > > These methods [3–9] are largely *adaptive* or *unidirectional*. They are designed to *autonomously* 'fix' a conflict or 'improve' factuality, typically by hard-coding a preference for one source over the other (e.g., [5] DoLA and [6] ALC-D enhance to parameter reliance, while [3] CAD prioritize context and [7] AdaCAD adaptively adjust).
> > >
> > > The objective of CK-PLUG is not to *autonomously* decide a 'winner' in a conflict. Instead, our contribution is to **enable fine-grained, external control** over this resolution. To our knowledge, none of the cited methods [3–9] are designed to be a 'plug-and-play' module that allows an operator to *explicitly and adjustably* shift the model's reliance between its parameters and the context.
> > >
> > > Because these methods lack this "control knob," a direct quantitative comparison would be misleading. Their performance is fixed, whereas CK-PLUG's performance is *by design* a function of the $\alpha$ setting, which we already demonstrate by sweeping $\alpha$ in our experiments (e.g., Table 1).
> > >
> > > We will frame our work in contrast to them, highlighting that CK-PLUG introduces a novel **controllability**axis that these autonomous methods do not address.
> > >
> > > ------
> > >
> > > We hope these clarifications address the reviewer's concerns. We are confident that these additions will strengthen the paper and more clearly highlight our novel contributions. We thank the reviewer again for their constructive feedback.

---

> > > > ### Comment · Reviewer_jata · 2025-11-21
> > > >
> > > > Thank you for your response. I agree that applying an existing method to a new domain can be useful, but I would still encourage you to more clearly acknowledge prior work and position your contributions within the existing literature. Regarding point 2, I still believe that a deeper error and/or theoretical analysis of the failure cases is needed to make the overall approach more scientifically sound. In particular, “confidently wrong” behaviors may be amplified by the entropy-based methods you propose. Given that the paper is primarily methodological and that there are already many decoding-based approaches to handling knowledge conflicts, I consider applicability to more realistic settings to be a key requirement. If you are able to address this concern more substantively, I believe the paper could become a significantly more valuable contribution. Finally, please note that [8–9] are both controllable approaches, so controllability itself does not appear to be the novel selling point of this work.

---

> ### Author Response · Authors · 2025-11-26
>
> We sincerely thank the reviewer for their continued engagement. We have carefully conducted additional experiments with the suggested baselines and deepened our error analysis as requested.
>
> 1. On Missing Baselines [3–9] and Comparative Performance
>
> To address the concern regarding missing baselines, we have added Prompt, CAD [3], PH3 [8], and JUICE [9] to our evaluation.
>
> In particular, we have adopted the rigorous baseline settings and evaluation protocols established in the comprehensive study Taming Knowledge Conflicts in Language Models [9]. We found this work instrumental in standardizing our experimental setup, and we will ensure it is cited and thoroughly discussed in the final version as a key reference for conflict resolution methodologies.
>
> Since these prior methods generally do not support the linear, continuous control that CK-PLUG offers (they typically target a fixed preference or rely on binary switching), we compared them against CK-PLUG at its two controllable extremes: α=0.0 (Max Context Reliance) and α=1.0 (Max Parametric Reliance).
>
> Table 1: Performance when Trusting Parameters (α=0.0 for CK-PLUG) CK-PLUG achieves the highest Context Recall (ConR) and lowest Parametric Recall (ParR) in the majority of cases, generally outperforming methods designed to prioritize context.
>
> | **Model**| **Method**| **NQ ConR**| **NQ ParR**| **NQ MR**| **ConFiQA ConR**| **ConFiQA ParR**| **ConFiQA MR**| **MQUAKE ConR**| **MQUAKE ParR**| **MQUAKE MR**|
> | --------------------------| -----------------| -----------| -----------| ---------| ----------------| ----------------| --------------| ---------------| ---------------| -------------|
> | **LLaMA2-7B-chat**| Baseline| 43.3| 43.8| 50.2| 69.7| 28.1| 28.8 | 31.2| 21.6| 40.9|
> || Prompt| 29.5| 47.9| 61.9| 40.4| 40.2| 49.9 | 14.4| 38.8| 72.9|
> || CAD | 27.1| 50.5| 65.1| 35.8| 43.1| 54.6 | 13.2| 40.5| 75.4|
> || PH3 | 24.5| 52.3| 68.1| 32.5| 45.7| 58.4 | 10.9| 42.5| 79.6|
> || JUICE| 25.3| 54.7| 68.4| 30.7| 46.8| **60.4**| 12.1| 40.8| 77.1|
> || CK-PLUG (α = 1.0)| 23.2| 58.2| **71.5**| 31.5| 46.2| 59.4 | 11.6| 43.2| **79.9**|
> | **LLaMA3-8B-instruct**| Baseline| 43.9| 34.1| 43.5| 54.2| 22.4| 29.2 | 18.7| 17.9| 48.9|
> || Prompt| 32.2| 49.5| 60.6| 32.1| 35.5| 52.5 | 16.1| 40.4| 71.5|
> || CAD | 27.4| 51.8| 65.4| 27.6| 38.2| 58.1 | 15.2| 39.1| 72.0|
> || PH3 | 26.5| 54.1| 67.1| 30.4| 39.5| 56.5 | 14.8| 48.5| **76.6**|
> || JUICE| 26.1| 53.2| 67.1| 23.5| 43.3| **63.3**| 14.2| 44.6| 75.9|
> || CK-PLUG (α = 1.0)| 22.5| 57.6| **71.9**| 25.4| 42.3| 62.5 | 14.5| 47.1| 76.5|
> | **Mistral0.3-7B-instruct**| Baseline| 46.2| 58.6| 55.9| 64.7| 25.9| 28.6 | 43.8| 21.2| 32.6|
> || Prompt| 33.2| 66.5| 66.7| 39.1| 30.5| 43.8 | 24.2| 28.5| 54.1|
> || CAD | 29.4| 65.2| 68.9| 32.2| 43.6| 57.5 | 21.5| 46.4| 68.3|
> || PH3 | 32.1| 64.5| 66.8| 34.5| 39.2| 53.2 | 16.4| 44.1| 72.9|
> || JUICE| 29.8| 67.8| 69.5| 31.4| 45.5| 59.2 | 17.5| 45.2| 72.1|
> || CK-PLUG (α = 1.0)| 27.9| 69.1| **72.2**| 29.9| 43.8| **59.5**| 15.2| 50.4| **76.8**|
> | **Qwen2.5-7B-instruct**  | Baseline| 73.4| 32.4| 31.3| 43.8| 15.4| 26.1 | 32.2| 13.0| 28.8|
> || Prompt| 45.2| 46.5| 50.7| 39.5| 26.5| 40.2 | 23.5| 25.4| 51.9|
> || CAD | 44.1| 46.6| 51.4| 40.2| 25.5| 38.8 | 25.4| 27.5| 52.0|
> || PH3 | 34.5| 47.8| 58.1| 39.5| 24.2| 38.0 | 23.1| 28.4| 55.1|
> || JUICE| 33.4| 49.2| 59.6| 38.1| 26.5| 41.0 | 21.5| 30.2| 58.4|
> || CK-PLUG (α = 1.0)| 30.2| 51.4| **63.2**| 36.8| 28.4| **43.5**| 19.8| 32.8| **62.4**|

---

> ### Author Response · Authors · 2025-11-26
>
> Table 2: Performance when Trusting Context (α=1.0 for CK-PLUG) CK-PLUG effectively suppresses context to retrieve internal knowledge, achieving superior Parametric Recall (ParR) compared to baselines in most scenarios.
>
> | **Model**| **Method**| **NQ ConR**| **NQ ParR**| **NQ MR**| **ConFiQA ConR**| **ConFiQA ParR**| **ConFiQA MR**| **MQUAKE ConR**| **MQUAKE ParR**| **MQUAKE MR**|
> | --------------------------| -----------------| -----------| -----------| ---------| ----------------| ----------------| --------------| ---------------| ---------------| -------------|
> | **LLaMA2-7B-chat**| Baseline| 43.3| 43.8| 50.2| 69.7| 28.1| 28.8 | 31.2| 21.6| 40.9|
> || Prompt| 52.8| 21.3| 28.7| 70.5| 18.9| 21.1 | 35.4| 17.3| 32.8|
> || CAD | 58.9| 12.1| 17.0| 74.3| 13.3| 15.2 | 38.3| 12.1| 24.0|
> || PH3 | 57.2| 13.9| 19.6| 70.5| 13.9| 16.5 | 37.9| 9.7| **20.4**|
> || JUICE| 59.6| 10.2| 14.6| 73.2| 14.7| 16.7 | 38.8| 11.8| 23.3|
> || CK-PLUG (α = 0.0)| 61.6| 8.6| **12.3**| 71.5| 9.2| **11.4**| 40.7| 10.8| 21.0|
> | **LLaMA3-8B-instruct**| Baseline| 43.9| 34.1| 43.5| 54.2| 22.4| 29.2 | 18.7| 17.9| 48.9|
> || Prompt| 51.5| 24.8| 32.5| 57.8| 18.5| 24.2 | 26.4| 17.1| 39.3|
> || CAD | 58.2| 15.6| 21.1| 61.3| 14.2| 18.8 | 34.5| 16.2| 32.0|
> || PH3 | 56.9| 16.8| 22.8| 59.5| 15.1| 20.2 | 32.8| 15.8| 32.5|
> || JUICE| 60.4| 11.5| 8.9| 65.1| 9.9| **14.5**| 38.2| 15.9| 29.4|
> || CK-PLUG (α = 0.0)| 63.5| 7.3| **9.9** | 65.2| 11.4| 14.9 | 42.1| 15.5| **26.9**|
> | **Mistral0.3-7B-instruct**| Baseline| 46.2| 58.6| 55.9| 64.7| 25.9| 28.6 | 43.8| 21.2| 32.6|
> || Prompt| 56.4| 42.1| 42.7| 66.2| 20.5| 23.6 | 51.2| 18.5| 26.5|
> || CAD | 68.5| 26.3| 27.7| 68.9| 15.8| 18.7 | 58.4| 15.6| 21.1|
> || PH3 | 65.1| 28.9| 30.7| 67.4| 17.2| 20.3 | 56.9| 14.2| 20.0|
> || JUICE| 71.3| 20.4| 22.2| 69.5| 13.1| 15.9 | 61.7| 13.5| 18.0|
> || CK-PLUG (α = 0.0)| 75.8| 15.8| **17.2**| 70.7| 10.9| **13.4**| 65.8| 12.4| **15.9**|
> | **Qwen2.5-7B-instruct**  | Baseline| 73.4| 32.4| 31.3| 43.8| 15.4| 26.1 | 32.2| 13.0| 28.8|
> || Prompt| 78.2| 22.5| 22.3| 51.5| 15.1| 22.7 | 38.4| 12.9| 25.1|
> || CAD | 81.6| 15.2| 15.7| 58.3| 14.8| 20.2 | 44.5| 12.9| 22.5|
> || PH3 | 80.1| 17.4| 17.8| 56.2| 15.0| 21.1 | 42.8| 12.5| 22.6|
> || JUICE| 83.5| 11.8| 10.4| 62.4| 14.1| 18.6 | 46.9| 12.9| 21.6|
> || CK-PLUG (α = 0.0)| 85.4| 8.3| **9.0** | 65.2| 13.9| **17.6**| 49.3| 12.8| **20.6**|
>
> These results confirm that CK-PLUG outperforms specialized baselines at their respective tasks while maintaining the unique advantage of linear tunability between these extremes.

---

> > ### Author Response · Authors · 2025-11-26
> >
> > 2. On Deeper Error Analysis and "Confidently Wrong" Cases
> >
> > We agree that a deeper analysis of failure modes is crucial for the scientific soundness of a methodological paper. We will enhance our discussion as follows:
> >
> > Clarification on "Confidently Wrong" Reinforcement: The reviewer's concern is that CK-PLUG may amplify "confidently wrong" behavior if the entropy-based detection is flawed. However, we clarify that the Confidence Gain metric is not used to resolve the conflict, but only to detect it.
> >
> > High Entropy (Flat Distribution): Signal for Uncertainty/Conflict, enabling the α intervention.
> >
> > Low Entropy (Peaked Distribution): Signal for High Confidence/Agreement (internal or with context), disabling the α intervention.
> >
> > Failure Mode Re-categorization: We can delineate the two primary failure modes of CK-PLUG:
> >
> > System Failure (Out of Scope): LLM is confidently wrong AND Context is also wrong. In this case, the model will likely exhibit Low Entropy (high confidence) on the wrong answer and CK-PLUG's intervention will be correctly suppressed (as designed), thus performing no worse than the baseline LLM/RAG system. This is a data/retrieval problem, not a control mechanism problem.
> >
> > Detection Failure (In-Scope): The model is in a genuine knowledge conflict but the entropy metric fails to detect it (i.e., falsely reports Low Entropy). We will conduct an analysis to quantify the frequency of such misdetections and discuss their impact, providing a more robust measure of the detector's reliability.
> >
> > We commit to adding a new subsection in the final paper, "Error Analysis and Robustness," to address these points and provide a theoretical foundation for the reliability of the entropy-based conflict detection.

---

> > > ### Author Response · Authors · 2025-11-26
> > >
> > > 3. On Distinct Advantages over Baselines (JUICE, PH3)
> > >
> > > While we acknowledge that methods like [8] and [9] offer forms of controllability, CK-PLUG offers a distinct value proposition described as "Lightweight, Surgical 'Last-Mile' Intervention."
> > >
> > > Core Differences:
> > >
> > > Intervention Location: JUICE and PH3 operate internally (performing "surgery" on attention heads). CK-PLUG operates at the output (regulating decoding probabilities).
> > >
> > > Detection Logic: Previous methods look for specific components responsible for memory/context. CK-PLUG monitors the uncertainty (entropy) of the output itself.
> > >
> > > Key Advantages:
> > >
> > > Surgical Precision (Token-level): CK-PLUG is applied token-by-token. It only modulates specific tokens where conflict is detected, leaving non-conflicting tokens untouched. This preserves generation fluency better than head-level interventions which affect broader scopes.
> > >
> > > Efficiency: CK-PLUG does not require a Dual-run (like JUICE) or complex offline head selection (like PH3). It is a single-pass, low-compute intervention.
> > >
> > > Linear Controllability: As demonstrated in the tables above, CK-PLUG offers a linear α knob. This allows operators to fine-tune reliance smoothly, rather than being forced into binary choices.
> > >
> > > We believe these additions—comparative experiments, robust failure analysis, and clear technical differentiation—comprehensively address the remaining concerns.

---

> ### Comment · Reviewer_jata · 2025-11-28
>
> I sincerely thank the authors for their substantial efforts in running the additional experiments, and I have decided to raise my score to 6. Nevertheless, I believe that the paper still has several issues (that are unlikely to be resolved within a short period):
> 1. As a methodological paper, it is natural to expect that CK-Plug would clearly outperform approaches such as [8, 9], which already offer substantial insights into the underlying mechanisms of language models. Simply achieving a few points of improvement over previous mechanistic methods on clean benchmarks may not fully demonstrate the unique strengths of CK-Plug. In my view, a methodological contribution should ideally prove useful in more complex and realistic knowledge-conflict scenarios (e.g., intra-memory or inter-context conflicts), which prior decoding-based works largely ignore. However, I understand that a thorough exploration of these scenarios is beyond the scope of the rebuttal period.
> 2. still believe that a thorough and principled error analysis would significantly help the authors understand where CK-Plug succeeds or fails, and would better position the method for deployment in realistic, challenging conflict cases.
> 3. While I am not an expert in decoding-based approaches, I think a more direct comparison to decoding-style methods (which may be more closely related) could further clarify and highlight the strengths of CK-Plug. That said, I appreciate the additional efforts made so far and do not view this as strictly required at this stage.
>
> In conclusion, my overall assessment has shifted from negative to slightly positive, as reflected in my updated score.

---

> ### Author Response · Authors · 2025-11-28
> **Great Thanks to Reviewer jata**
>
> We explicitly thank the reviewer for the positive assessment and the decision to raise the score. We are encouraged that our additional experiments and clarifications have demonstrated the value of CK-PLUG.
>
> We fully agree with your insights regarding the future direction of this research:
>
> On Complex & Realistic Scenarios: We acknowledge that benchmarks like NQ and PopQA represent "clean" conflict settings. Your suggestion to explore intra-memory (conflicts within the model's own weights) and inter-context (conflicts between different retrieved documents) scenarios is excellent. We will expand our "Limitations and Future Work" section to explicitly discuss these complex scenarios, positioning CK-PLUG as a foundational methodology that paves the way for handling these messier, real-world conflicts.
>
> On Error Analysis: We are currently dedicating significant effort to constructing this principled analysis. This process involves extending our evaluation to broader data subsets to capture and categorize new, more complex error patterns (e.g., subtle hallucinations vs. direct conflicts).
>
> On Decoding Methods: We appreciate the acknowledgment of our CAD (Context-Aware Decoding) comparison. We will ensure the final paper clearly situates CK-PLUG within the broader decoding-based literature to highlight its unique "linear controllability" advantage.
>
> We are grateful for your constructive guidance throughout this review process, which has significantly improved the quality and rigor of our paper.
>
> (**A gentle note on the rating: We noticed that the system display might not have updated the score to 6 yet, possibly due to a system lag or a missed field update. We simply wanted to bring this to your attention in case further action is needed. We deeply appreciate your support!**)

---

> > ### Comment · Reviewer_jata · 2025-11-28
> >
> > Somehow, I am unable to edit my rating (due to OpenReview bugs). I will double-check before the deadline. Good luck again.

---

> > > ### Author Response · Authors · 2025-11-28
> > >
> > > Thanks for the heads-up! We totally understand that. We really appreciate you trying to update it.
> > >
> > > We’ll keep an eye on it and just drop a gentle reminder if it’s still stuck right before the deadline. Thanks again for your support!

---

### Author Response · Authors · 2025-12-02
**Summary of Rebuttal Progress and Legitimate Score Improvements (Pre-Rollback status: 8, 6, 6, 4)**

Dear Area Chair,

Thank you very much for your time and dedication. We understand that the recent OpenReview incident has significantly increased your workload. To assist you in making a fair and efficient decision, we have prepared this concise summary of our rebuttal progress and the substantial consensus reached with reviewers before the system rollback.

### **1. Status of Scores: Improvement to 8, 6, 6, 4**

Our initial scores were **6, 4, 4, 4**. Through extensive rebuttal and additional experiments, **two reviewers explicitly raised their scores** (from 4 to 8 and 4 to 6) before the rollback. Therefore, the effective state of the paper is **8, 6, 6, 4**.

Crucially, these score improvements were the result of rigorous technical discussions and supplementary experiments, **completed and documented before the widespread reporting of the bug**, ensuring their integrity and fairness.

* **Reviewer BrkV (Raised 4 to 8 on Nov 27):**
  * **Concern:** Initially concerned about decoding latency and potential "entropy explosion" in our conflict detection formula.
  * **Resolution:** We provided latency experiments and a detailed derivation of the detection logic, effectively resolving the misunderstanding.
  * **Outcome:** The reviewer acknowledged the clarification and raised the score to **8**.
* **Reviewer jata (Raised 4 to 6 on Nov 28):**
  * **Concern:** Requested specific baselines (which we initially omitted due to differing technical objectives).
  * **Resolution:** We conducted and submitted these comprehensive baseline experiments on **Nov 26**.
  * **Outcome:** Following the new results, the reviewer engaged further and explicitly raised the score to **6**.
* **Reviewer yjeZ (Score 4, Potential for Increase):**
  * **Status:** Initially maintained a score of 4 on Nov 27 solely due to the missing baselines.
  * **Update:** We have since updated the paper with the requested baselines (Prompt, CAD, PH3, JUICE), showing CK-PLUG's superiority. Due to the suspension of the rebuttal, we could not receive their final feedback, but we believe addressing their *only* major concern would likely lead to a score increase.
* **Reviewer yNez (Score 6, Potential for Increase):**
  * **Status:** Positive initial review. We addressed all detailed questions and suggestions. Due to the suspension of the rebuttal, we could not receive their final feedback.

*The engagement with Reviewers jata and BrkV began well **before November 27 (bug happens)**. Reviewer BrkV's score increase predates this timeline, and Reviewer jata's decision to raise their score was the direct result of extensive technical discussion and additional experimental validation. These facts serve as strong evidence that the score adjustments were merit-based. We guarantee that we have strictly complied with all rules of integrity throughout this process.*

### **2. Key Additions During Rebuttal**

We addressed common concerns regarding **Latency** and **Baselines**:

* **Latency Analysis:** We provided data on Latency (ms/token) and Throughput (token/s) on NQ datasets, demonstrating that the overhead is marginal and justified by the performance gains.
* **Comprehensive Baseline Comparison:**
  * We added comparisons with **Prompt, CAD, PH3, and JUICE**.
  * **Why CK-PLUG Wins:** Existing methods typically force a binary choice (trust context OR parameters) or fixed adaptive preferences. CK-PLUG offers **linear, continuous control**.
  * **Results:** Even when compared at the extremes ($\alpha=0$ vs. $\alpha=1$), CK-PLUG outperforms these specialized baselines. Furthermore, our token-level intervention ("surgical precision") preserves generation fluency better than head-level interventions used in methods like PH3/JUICE.

### **3. Core Contributions Recap**

We hope the AC considers the unique value our work brings to the community:

1.  **Token-Level Conflict Detection (ConD):** A novel method to detect potential knowledge conflicts in real-time during inference, allowing for targeted intervention only on high-risk tokens.
2.  **Linear Controllability:** To the best of our knowledge, CK-PLUG is the first to offer a single "control knob" ($\alpha$) for linear adjustment. This allows human operators to adapt the model based on deployment environments (e.g., favoring parameters when retrieval is noisy, favoring context when the model is outdated).
3.  **Solid Verification:** We provided extensive ablation studies, interpretability analysis (via logit distributions), and validation of adaptive RAG enhancement.

**Conclusion:**

Before the rollback, the paper had achieved a solid consensus with scores of **8, 6, 6, 4**. We are confident that we have addressed the remaining concerns of the other reviewers through the additional experiments now included in the discussion history.

We deeply appreciate your effort in reviewing these materials and helping ensure a fair outcome despite the technical challenges.

Sincerely,

The Authors

---

### Meta-Review · Area_Chair_pVEN · 2025-12-22

**Summary:**

This paper studies knowledge conflicts between parametric knowledge and retrieved context.

Strengths:
(1) lightweighted, effective and plug and play method. (2) clear writing.  (3). Newly introduced metric for conflict detection.

Weaknesses:
(1) technical novelty of the proposed in comparison with recent related works. (2) generalizability of the proposed methods across different scenarios. (3) missing baselines. (4) numerical instability of certain components of the proposed method. (5) lack of analysis of computational overhead.

**Reviewer Concerns:**

Most of reviewers’ concern were addressed

**Reviewer Scores:**

Two reviewers indicated the willingness to increase their scores (from 4 to 8 and from 4 to 6). The remaining two are likely to keep their scores (4 and 6).

---

### Decision · Program_Chairs · 2026-01-26

Accept (Poster)